# Thermoplasmonic neural chip platform for in situ manipulation of neuronal connections in vitro

Nari Hong [1] & Yoonkey Nam [1,2✉]

Cultured neuronal networks with a controlled structure have been widely studied as an in vitro model system to investigate the relationship between network structure and function. However, most cell culture techniques lack the ability to control network structures during cell cultivation, making it difficult to assess functional changes induced by specific structural changes. In this study, we present an in situ manipulation platform based on gold-nanorod-mediated thermoplasmonics to interrogate an in vitro network model. We find that it is possible to induce new neurite outgrowths, eliminate interconnecting neurites, and estimate functional relationships in matured neuronal networks. This method is expected to be useful for studying functional dynamics of neural networks under controlled structural changes.

[1] Department of Bio and Brain Engineering, Korea Advanced Institute of Science and Technology (KAIST), Daejeon 34141, Republic of Korea. [2] KAIST Institute for Health Science and Technology, Korea Advanced Institute of Science and Technology (KAIST), Daejeon 34141, Republic of Korea. ✉email: ynam@kaist.ac.kr

Unraveling how functional features of neural networks emerge from the underlying structural circuitry has been one of the central questions in the neuroscience field. To explore the relationship between structure and function, dissociated neuronal culture systems have been studied by taking advantage of their high accessibility and ease of manipulation[1]. Previous studies showed that engineered networks with controlled structure can be built by giving chemical or physical constraints in neuronal adhesion and outgrowth[2]. In chemical based approaches, using cell-attractive or repulsive materials, microfabrication technologies have been applied for organizing chemical cues spatially on a culture substrate to control the polarization of neurons and confine the morphology of a cultured network: photolithographic etching[3,4], lift-off[5,6], and micro-contact printing[7–10]. Alternatively, neurons were seeded onto microfabricated structures of biocompatible materials such as microwells and grooves for cell placement and neurite guidance, respectively, made by photolithography[11], soft-lithography[12], or micro-molding in capillaries[13]. Several types of microfluidic devices[14,15] or stencils[16,17] were also used for physical patterning methods.

Despite advances in in vitro neuronal culture technology that can control network structures, it is still challenging to identify functional changes in a network induced by alteration of structural connections. This was because most of the patterning techniques were static in the sense that they could create patterns on the substrate only before cell seeding and could not change once the cultivation started. Therefore, once neurons form a confined network on the engineered substrate relatively early in time, it is difficult to manipulate their connectivity by the traditional patterning methods. In terms of studying network functions, many studies have focused on investigating changes in electrical signals that occur while a network matures over several weeks in a culture dish, and it was found that there was a strong correlation between the overall structural changes and electrical activities of individual neurons in a network. However, it is difficult to track changes in a specific structural connection and the following changes in network activity. Thus, a new technological approach is needed to more directly investigate the impact of network structure on its function. It should be able to specifically induce structural changes in functionally matured networks and study the functional changes that accompany them. In particular, after realizing a designed neural network through an existing patterning method, it is desirable to manipulate physical connections to achieve both structural design and control.

To construct an in vitro model for studying network dynamics by manipulating structural changes during the development or maturation period, an in situ manipulation technique is needed that can drive changes in networks already formed, such as creating or removing the connections between networks at the desired locations and time points. Recently, several techniques based on a light-mediated approach were proposed to modify pre-constructed patterns in an in situ manner: a photochemical process[18,19], photoablation[20], photothermal etching[21,22], and photocavitation[23]. Although such developed methods allowed neurite guidance and connection at desired time points after positioning neurons, previous studies have focused on morphological changes at the cellular level for a short period of time and little effort has been made to extend the technique to investigate the functional changes at the network level.

The goal of this study is to develop a technological platform for in situ manipulation techniques to study the structure and function of neuronal networks. This platform should be able to induce structural changes at a desired time point during neuronal development and maturation, and investigate network connectivity affected by those changes. Our proposed platform is composed of a micropatterned hydrogel layer, near-infrared (NIR)-sensitive gold nanorod (GNR) layer, and a microelectrode array (MEA). The agarose hydrogel layer serves as a highly efficient cell-repulsive material for cell patterning, and as a thermosensitive material for in situ neurite manipulation. The thermoplasmonic GNR layer, which is the key component of the platform, delivers localized heat to the agarose hydrogel layer or neurons to create new connections by guiding neurite outgrowth, to remove existing connections by ablating neurites, and to modulate neural activity by suppressing spiking activity. The MEA enables spatiotemporal neural recording to analyze functional connectivity in a long-term period. In this study, micropatterns of the cell-repulsive agarose hydrogel were fabricated on the MEA to obtain confined neuronal networks that last for a month. We optimized the thermoplasmonic ablation of agarose hydrogel to create new neuronal connections between the networks, and successfully guided neurite outgrowth from the matured neuronal networks. Thermoplasmonic ablation of the neuronal processes was also successful to eliminate the existing connections between synchronized networks. Finally, we could modulate the network activity to map a functional connectivity of newly connected networks by thermoplasmonic neural inhibition. During the in situ manipulation, we could track and identify the changes in functional connectivity of networks for a few weeks.

## Results

### Design and fabrication of the thermoplasmonic interface.
Figure 1a shows a schematic illustration of our platform incorporating a thermoplasmonic interface and a thermosensitive hydrogel onto an MEA for in situ manipulation and monitoring of neuronal networks. The NIR-sensitive GNR layer was located on the MEA surface so that localized surface plasmon resonance could deliver thermoplasmonic heat directly to the thermosensitive agarose hydrogel or neurons. Three different manipulations could be implemented on the platform. First, we can locally ablate the agarose hydrogel and induce new neurite outgrowth (Fig. 1b–1). By applying such manipulation of hydrogel ablation to matured networks that exhibit spontaneous network activities, we can examine the change in functional connectivity as two or three networks become structurally connected. Second, local heating from the GNRs can be used for neurite ablation (Fig. 1b–2). By ablating neural connections selectively, we can investigate how the network activity and connectivity changes within and between networks. Third, we can modulate network activity by delivering photothermal stimulation to firing neurons to map the functional connectivity of newly connected networks (Fig. 1b–3). The switching between ablation and modulation was achieved by the input NIR laser power: higher levels for ablation and lower levels for modulation.

To fabricate the platform, we first synthesized GNRs with a longitudinal plasmon resonance peak in the NIR region and formed a thermoplasmonic GNR layer on the MEA through electrostatic binding (Fig. 1c). As GNRs were tuned to have a maximum absorption peak in NIR region (789 nm), the thermoplasmonic interface could generate localized heat at the position where the NIR laser (785 or 808 nm) was focused. After coating a cell-adhesive molecule, poly-D-lysine (PDL), the thermosensitive agarose hydrogel was patterned on the GNR layer to control the neuronal attachment (Fig. 1d, e).

### Optimization of thermoplasmonic ablation of agarose hydrogel and effect of laser power on sprouting neurite outgrowth.
The thermosensitive agarose hydrogel (melting temperature: 87–89 °C) on the GNR layer could be successfully ablated by the

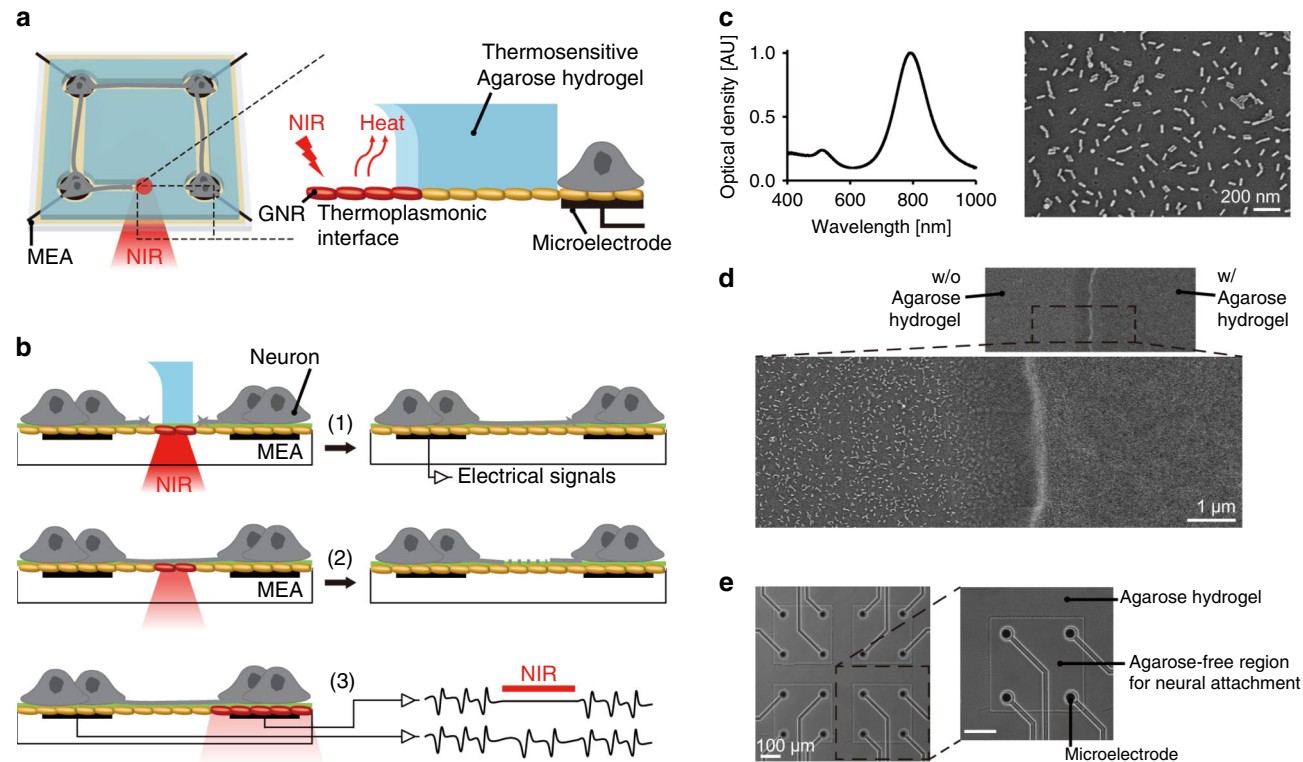

**Fig. 1 Designed platform and chip characterization. a** Schematics of the developed platform using gold nanorods (GNRs), agarose hydrogel, and a microelectrode array (MEA). **b** Overview of experimental design for in situ manipulation. (1) Thermoplasmonic hydrogel ablation to create new connections. (2) Thermoplasmonic neurite ablation to remove existing connections. (3) Thermoplasmonic modulation to map network connectivity. **c** Absorbance spectrum of GNRs and SEM image of the immobilized GNRs on a substrate. **d** SEM images of patterned agarose hydrogel (air-dried) on the GNR-coated substrate. **e** Phase-contrast images of agarose hydrogel patterns on the GNR-coated MEA. The images in **c**–**d** and **e** are representative of two and more than ten replicates, respectively.

generated heat from GNRs, and the ablation width could be controlled by varying the illuminated laser power density. The thermoplasmonic stimulation was applied in a power density range of 30–210 W/mm$^2$ (10–69 mW, 785 nm, beam diameter: 20.5 μm) while moving the substrate in one direction to ablate the agarose into the shape of a line (Supplementary Fig. 1). As shown in phase-contrast images of Fig. 2a, the agarose hydrogel was ablated for power densities higher than 60 W/mm$^2$ and the width increased with increasing laser power density. The width is plotted against the laser power density in Fig. 2b. At 60–210 W/mm$^2$, the hydrogel was completely ablated and the width of ablated lines increased from 10 to 46 μm, showing a strong linear relationship ($R^2 = 0.9972$). In this range, the 10 μm-thick agarose layer was removed, forming open micro-sized channels (Supplementary Fig. 2). As we tuned the longitudinal absorption peak of GNRs (789 nm) close to the wavelength of the laser that was applied (785 nm), we could achieve high photothermal conversion efficiency for agarose ablation (Supplementary Fig. 3).

Although the GNR layer could generate sufficiently high heat to ablate the agarose hydrogel, the adhesive molecule for neuron attachment below the removed hydrogel was not damaged. To determine whether the heat denatures a biomolecule, laminin was coated before agarose patterning, and it was stained by immunostaining after the ablation process (Fig. 2a). Up to 150 W/mm$^2$, the laminin in ablated regions was not denatured, whereas noticeable partial damage of laminin at the center of the ablated region was observed at more than 180 W/mm$^2$.

After the ablation process, the neurite outgrowth along the ablated regions was examined to determine the range of laser power density for creating neurite connections. At 6 days in vitro

(DIV), the agarose hydrogel was ablated in a linear shape with 500 μm length using different power density. The neurite outgrowth was then quantified at 13 DIV, which was 1 week after the agarose ablation. In Fig. 2c, the axons extended through the ablated line using a power density between 90 and 210 W/mm$^2$ without cell body entering. At 60 W/mm$^2$, neurites did not enter the narrow-ablated line, and beyond 180 W/mm$^2$ where laminin was partially damaged, the extending axons tended to grow along the edge of the lane. Based on these results, a laser power density ranging from 90 to 150 W/mm$^2$ was used to ablate agarose hydrogel in further experiments for in situ neurite manipulation.

**Effect of new connections on network synchronization.** To generate new neurite connections between matured neuronal networks and investigate their impact on functional change in network connectivity, we designed a modular network using two unconnected networks ('modules') on an MEA and applied thermoplasmonic ablation to establish biological interconnections between two networks. As shown in Fig. 3a, each agarose well pattern contained four microelectrodes and neurons within the well constituted an individual network ($N_1$ and $N_2$). At 16 DIV, agarose hydrogel between two wells, marked with white dash lines (100 μm in length), was removed by ablating with the laser power density of 136 W/mm$^2$ (45 mW) (Fig. 3b–0 d). After the ablation process, neurites started to grow out from both wells and extended along the interconnection line within 24 h (Figs. 3b–1 d). Growth cones of outgrowing neurites could be spotted during this time (white arrowheads). The neurites reached the opposite well

## a

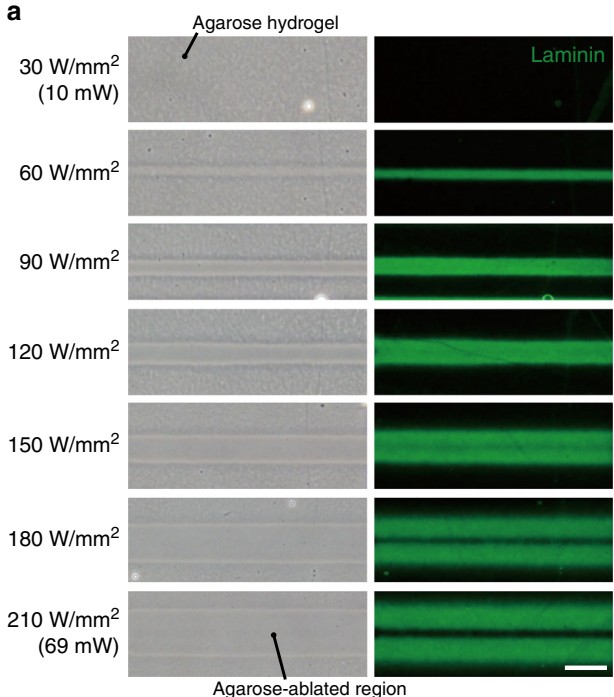

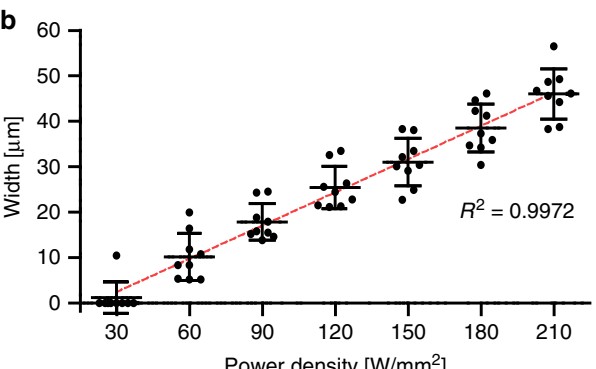

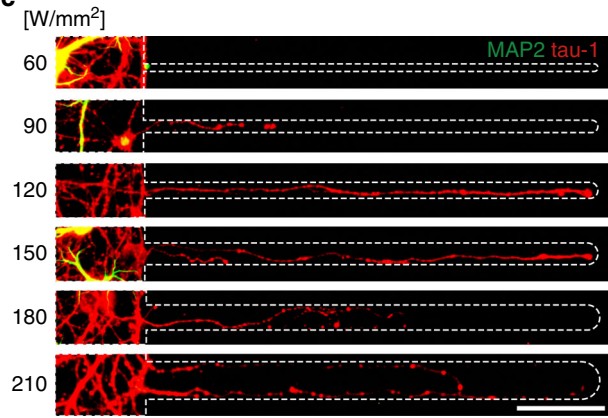

**Fig. 2 Thermoplasmonic ablation of agarose hydrogel and neurite outgrowth. a** Phase-contrast and anti-laminin staining images of ablated regions with different laser power densities. The green region indicates intact laminin, which was originally below the hydrogel and then revealed after agarose ablation. Scale bar: 50 μm. The images shown are representative of three independent experiments. **b** Ablated width of agarose hydrogel against power density. $n = 9$ ablated lines. Data are presented as mean ± SD. **c** Immunostained images of extended neurites along the ablated lines (green: MAP2, somatodendritic marker; red: tau-1, axonal marker). Scale bar: 100 μm. The images shown are representative of two independent experiments. Source data are provided as a Source Data file.

from four electrodes (Fig. 3a, Ch1 and C2 in $N_1$, Ch3 and Ch4 in $N_2$) and Fig. 3d presents raster plots of spike trains before the ablation and 3, 5, and 7 days after the ablation. After connecting two networks ($N_1$ and $N_2$), synchronized firing events between two networks emerged. To quantify the change in functional connectivity within or between networks, correlation coefficients between all electrode pairs were calculated. As shown in the correlation matrices (Fig. 3e), neurons formed local circuits in each network before the ablation, and spikes were only correlated between electrodes from same networks ($N_1$–$N_1$, $N_2$–$N_2$, $N_3$–$N_3$), while there was no correlation between networks ($N_1$–$N_2$, $N_1$–$N_3$, $N_2$–$N_3$). The connected networks ($N_1$–$N_2$) gradually became functionally synchronized between 3 and 7 days after the ablation (after 3 days: −0.002, after 5 days: 0.273, after 7 days: 0.460) and the correlation values reached close to those from local circuits (after 7 days; $N_1$–$N_1$, $N_2$–$N_2$: 0.660). Figure 3f shows the mean coefficient values from four different two-network samples that were connected after 2 weeks through one or two interconnection lines. The intra-correlation (intra-CC), which is the measure of the local synchrony in each module, had high coefficient values both before and after the ablation. This indicated that neurons in each network had established their own functional connections such that they tend to fire coincidently even after new connections are formed with the other network (mean intra-CC; before: 0.613, after 7 days: 0.775). In the case of the inter-correlation coefficients (inter-CC), which measures the functional synchrony of connected modules, the correlation values markedly increased after 2–6 days from the ablation. The increase of inter-CC reflected that neurons from different networks had a tendency to participate in the globally synchronized events (SEs). The inter-CC from unconnected networks remained close to zero for 7 days. Similar trends were also observed in larger networks connected through various numbers of interconnection lines (Supplementary Fig. 4). Taking together, the results demonstrate that we successfully induced structural changes by adding new neurite connections between the matured networks using thermoplasmonic ablation of agarose hydrogel, and traced the emergence of functional synchronization in connected networks over several weeks in situ.

**Thermoplasmonic neural ablation and network desynchronization.** Next, we attempted neurite ablation using thermoplasmonic stimulation and applied this technique to change the network structure by selectively removing biological connections in a modular network. First, we examined the morphological effects of thermoplasmonic ablation on neurites using optical micrographs. To perform the micro-ablation of neurites, neurite outgrowth was induced in interconnection lines at 6 DIV using the same method as described in the previous section. After 7 days of neurite outgrowth, the NIR laser was focused on the center of neurites to induce thermoplasmonic neurite ablation

within 1–2 days after ablation (Figs. 3b–2d), and more neurites grew through this interconnection line over time. As a result, we succeeded in forming new structural connections between two matured networks that were cultured separately for about 2 weeks.

After creating new neurite connections, the activities of connected networks were investigated. Figure 3c shows representative neuronal signals recorded before and after the ablation

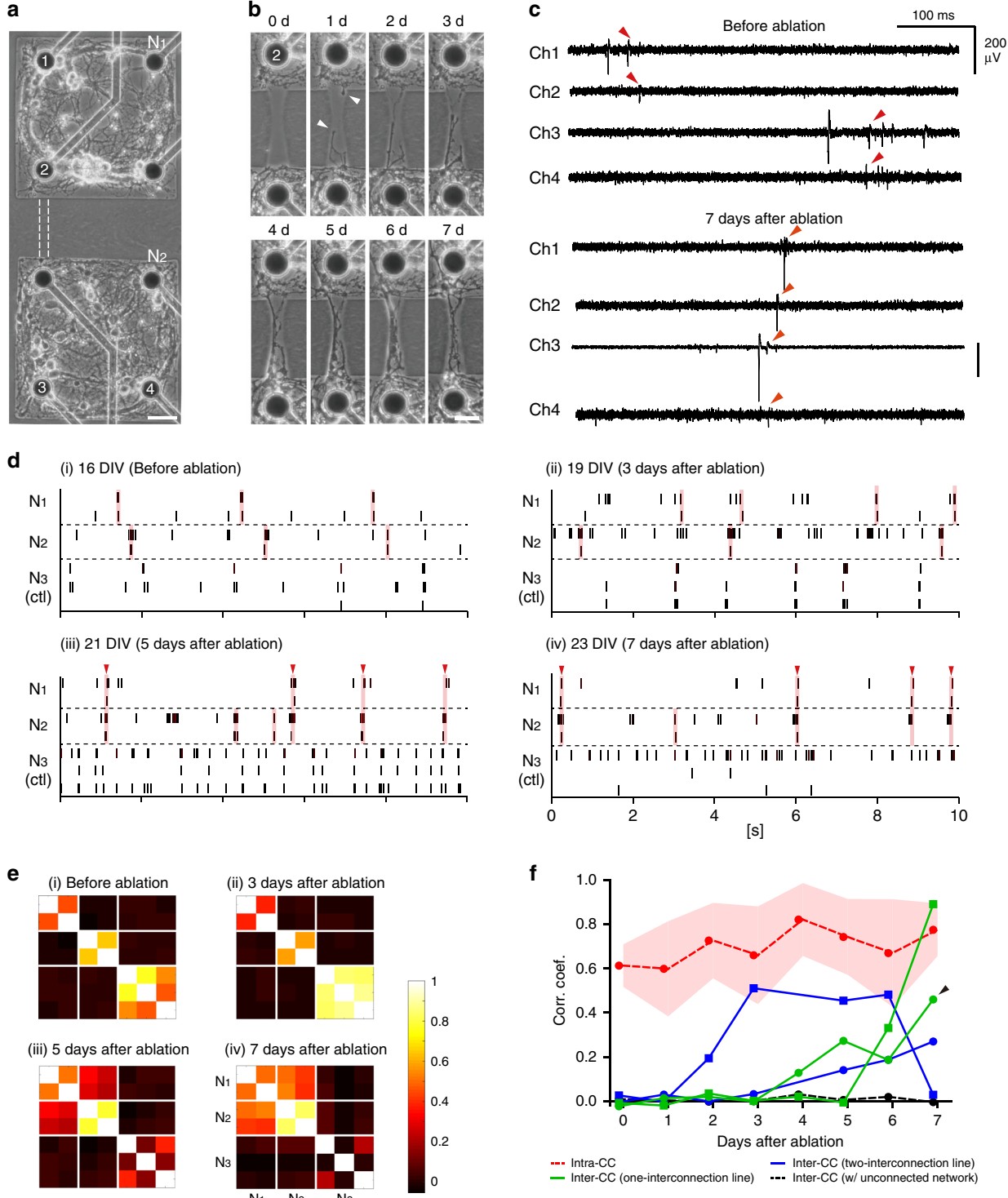

**Fig. 3 Network synchronization by creating new neurite connections. a** Phase-contrast image of neuronal networks at 16 DIV. Agarose micropatterns were 300 μm by 300 μm with a spacing of 100 μm. Scale bar: 50 μm. The image shown is representative of four independent experiments. **b** Neurite outgrowth into the interconnection line over time (d: days after ablation). Scale bar: 30 μm. **c** Raw traces recorded from four electrodes (Ch1 and Ch2 in $N_1$, Ch3 and Ch4 in $N_2$) before and 7 days after agarose ablation. Arrowheads indicate synchronized activities. **d** Raster plots before and 3, 5, 7 days after ablation. Pink shaded regions indicate synchronized activities within a 100 ms window. Red arrowheads indicate synchronized activities across $N_1$ and $N_2$. **e** Representative correlation matrices before and 3, 5, 7 days after ablation process. **f** Mean correlation coefficients of four MEAs. Green and blue graphs show inter-CC between networks with one and two-interconnection lines, respectively, and each line represents an individual MEA. An arrowhead indicates the values of the MEA in Fig. 3e. The red graph and shaded region denote the mean and SD values of intra-CC from 4 MEAs, respectively. Source data are provided as a Source Data file.

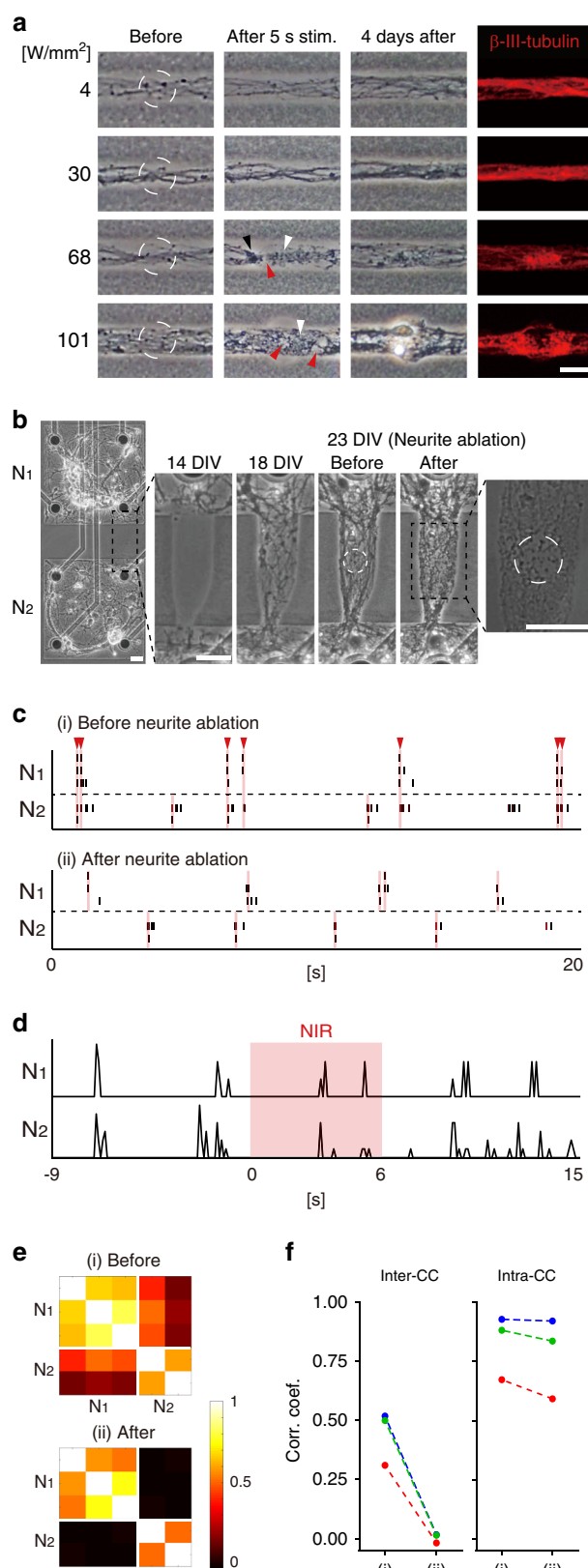

**Fig. 4 Thermoplasmonic neurite ablation and desynchronization between disconnected networks. a** Phase-contrast and anti-beta-III-tubulin staining images of ablated neurites. Red, black, and white arrowheads indicate neurite transection, beading, and fragmentation, respectively. Scale bar: 20 μm. The images shown are representative of two independent experiments. **b** Phase-contrast images of neurite outgrowth after agarose ablation (14 DIV, 18 DIV) and before and after neurite ablation (23 DIV). The NIR-focused area is shown by a white circle. Scale bar: 30 μm. The images shown are representative of three independent experiments. **c** Raster plots of electrodes in $N_1$ and $N_2$ before and after ablation. Pink shaded regions indicate synchronized activities within a 100 ms window. Red arrowheads indicate synchronized activities across $N_1$ and $N_2$. **d** Array-wide rate histograms (100 ms bins) before, during (pink shade), and after stimulation. **e** Representative correlation matrices before and after stimulation. **f** Mean correlation coefficients of three MEAs before (i) and after (ii) neurite ablation. Each graph indicates an individual MEA. The red graph shows the values of the MEA in Fig. 4e.

(19 mW)[24]. At the power intensities of 4 and 30 W/mm², there was little change of the neurite morphology (Fig. 4a). At power densities of 68 and 101 W/mm², however, morphological neurite alterations occurred in and around the NIR-focused region. The observed morphological changes were neurite transection, beading, and fragmentation, which were used as indicators of neurite injury and degeneration in cultured nerve cells[25–27]. At the highest power level of 101 W/mm², it was found that the surrounding agarose hydrogel also melted due to the thermoplasmonic heat. This strongly implied that the observed neurite damage and ablation were due to thermoplasmonic heating. The fluence of laser irradiation for neurite ablation (68–101 W/mm² for 5 s, 340–505 J/mm²) was higher than in previous studies (24–48[28] or 60–720[29] or 18–54 J/mm² [30]), in which in vitro photothermal ablation of cancer cells was performed using GNR membrane binding or uptake. Another interesting observation was that there was extended outgrowth of neurites over the ablated areas after 4 days from the neurite ablation. It can be inferred that even after removing the connections between the networks through the thermoplasmonic ablation, neurites can continue to grow out from the damaged or non-damaged neurons, thereby creating new connections again. Thus, it was found that thermoplasmonic stimulation could temporarily induce micro-sized local damage of neurites for structural change, and neurites could regrow over the stimulated area.

Next, we investigated the effect of thermoplasmonic neural ablation on the electrical activity of a synchronized neuronal network. To obtain a synchronized network, separated neuronal networks were patterned on an MEA and they were connected by the agarose hydrogel ablation method at 14 DIV (Fig. 4b). At 23 DIV, the connected networks were well-synchronized with each other, showing coincident spiking activities (pink shaded regions with red arrowheads in Fig. 4c (i)). Thermoplasmonic stimulation with 68 W/mm² was locally applied for 6 s to ablate interconnecting neurites (Fig. 4b). Fragmented neurites found both in and around the laser-focused region confirmed the successful neurite ablation. After the stimulation, the synchronous spiking activities between two modules disappeared and only local activity within individual modules appeared to remain (Fig. 4c (ii)). The array-wide rate histograms of two networks showed that the desynchronization between two modules occurred immediately after NIR illumination (Fig. 4d). Furthermore, the network synchrony between two modules ($N_1$–$N_2$) dramatically declined to nearly zero (Fig. 4e). As shown in Fig. 4f, the inter-CC decreased in three different samples (from 0.310 to −0.018; from 0.499 to 0.015; from 0.519 to 0.016), while the intra-CC remained

(Fig. 4a, white dashed circles). The focused NIR laser (785 nm, beam diameter: 20.5 μm) with 4, 30, 68, and 101 W/mm² (1, 10, 22, and 33 mW) was illuminated for 5 s. This power range was determined based on a previous study on the effect of a thermoplasmonic laser power on neural spiking activity, which showed irreversible loss of neural activity at 60.8 W/mm²

high (from 0.673 to 0.592; from 0.882 to 0.836; from 0.929 to 0.921). This indicated that individual networks still maintained the synchronized activities within themselves despite the removal of overall synchrony between networks. Thus, we concluded that the thermoplasmonic neurite ablation was sufficient to eliminate functional connections between synchronized networks, and this method could be used to control structural changes to study functional connectivity in neuronal networks.

**Network mapping by thermoplasmonic neuromodulation.** Finally, we show that the network connectivity of engineered modular networks can be investigated by the proposed thermoplasmonic platform at lower input power density levels. Based on our successful manipulation of network connections and observation of emergent synchronization, we attempted to estimate functional influence or dependency between network modules using our platform. According to previous studies, it is possible to suppress neural activity by delivering heat to neurons using thermoplasmonic gold nanoparticles[31]. Unlike electrical stimulation or optogenetics, thermoplasmonic neural stimulation can reversibly suppress neural activity at a desired scale by controlling the light power and does not require genetic modification, which would be beneficial for mapping network connectivity in situ. To measure the functional influence between network modules, the spiking activity in one module ('network $N_1$') was completely suppressed by thermoplasmonic NIR stimulation (TP-NIR) and the corresponding change from another module ('network $N_2$') was evaluated (Fig. 5a). The network-to-network influence of network $N_1$ on $N_2$ ($I_{N1 \rightarrow N2}$) was defined as the baseline activity change in the network $N_2$ at the maximum suppression of the modulating network $N_1$.

The first example shows a synchronized modular network composed of two modules (Fig. 5b). Two networks in agarose micropatterns were connected with one interconnection line by ablating the hydrogel at 15 DIV, and they became synchronized from 22 DIV (Supplementary Fig. 4). A functional influence test was executed at 34 DIV when the synchrony measure was relatively high (inter-CC: 0.407). To eliminate the network activity completely, the NIR illumination pattern was designed to cover the whole area of one network module. The laser power densities for the modulation were much lower than those used for the ablations (30–243 mW/mm² vs. 68–150 W/mm²). As shown in representative raster plots and perievent histograms (Fig. 5c, d), modulating one network induced activity changes in the other network. When $N_1$ was fully suppressed (TP-NIR $N_1$, Ch1, 243 mW/mm²), the spiking activity in $N_2$ (Ch2) appeared to be perturbed. When $N_2$ was suppressed (TP-NIR $N_2$, Ch2, 140 mW/mm²), the spiking activity in $N_1$ (Ch1) activity completely vanished, which implies that the $N_1$ spiking activity was highly influenced by $N_2$. Figure 5e, f show the spike rate change following the thermoplasmonic modulation in each network. Under the full inhibition of the modulated network, there were significant baseline activity changes for both cases (TP-NIR $N_1$ and TP-NIR $N_2$) compared to the spike changes in the control groups that were not connected. When $N_1$ was modulated from 0 to −97% (Fig. 5e), $N_2$ activity changed from 0 to −62%. When $N_2$ was modulated from 0 to −99% (Fig. 5f), $N_1$ activity changed accordingly (0 to −100%). The network-to-network influence was found to be unbalanced (Fig. 5h, $I_{N1 \rightarrow N2}$: 62%; $I_{N2 \rightarrow N1}$: 100%). In other words, there was an asymmetrical dependency between two networks. Spiking activity of $N_1$ was 100% driven by external connections from $N_2$, while some portion of the activity in $N_2$ (62%) was driven by external connections and the remaining portion was driven by the local recurrent circuits in $N_2$. This implies that functional connectivity was established in both

directions, but the degree of connectivity was stronger in $N_2 \rightarrow N_1$. To confirm our interpretation, we analyzed spatiotemporal patterns in spontaneous activity recordings. As shown in Fig. 5g, both network modules had bursting activity patterns. Among these bursts, some were associated with inter-network SEs and others were unassociated intra-network events. The percentage of total spikes participating in inter-network SEs were 69% and 38% for $N_1$ and $N_2$, respectively (Fig. 5i), which indicated that there was a large difference in the degrees of association with the global network activity. Furthermore, the spike propagation within the inter-network SEs was dominantly from $N_2$ to $N_1$ (Fig. 5j). In this synchronized network, functional mapping using the network-to-network influence measure was consistent with the functional analysis of spontaneous activity.

Next, we extended our analysis to a three-module network that was strongly synchronized. Three modular networks were connected with two interconnection lines at 21 days (Fig. 6a). Seventeen days after connecting three modules (38 DIV), the network exhibited strong synchronized bursting patterns (Fig. 6b). In each module, spikes from multiple electrodes showed a high degree of synchrony (Fig. 6c, mean intra-CC $N_1$: 0.839, $N_2$: 0.851, $N_3$: 0.858). For the three modules, inter-network synchrony measures showed that they were highly synchronized among each other (Fig. 6c and Supplementary Fig. 5, mean inter-CC: 0.576, $N_1$-$N_2$: 0.513, $N_1$-$N_3$: 0.455, $N_2$-$N_3$: 0.806), and nearly all bursts were highly associated with inter-network SEs. (Fig. 6d, $N_1$: 74.2%, $N_2$: 99.8%, $N_3$: 90.0%). Among the SEs, two persistent activity patterns dominated the network-wide activity: $N_3 \rightarrow N_2$ (71%) and $N_3 \rightarrow N_2 \rightarrow N_1$ (26%) (Fig. 6e). By thermoplasmonic modulation (Fig. 6f), nearly all of the spikes in the modulated network were suppressed at the highest power density. The network-to-network influence measure revealed that there was only one significant connection pair in this network (Fig. 6g, $I_{N3 \rightarrow N2}$: 45%), which corresponded to the persistent activity patterns found from spontaneous activity (Fig. 6e, $N_3 \rightarrow N_2$). We further analyzed the network connectivity using electrical stimulation and compared it with our influence measure. Figure 6h and i show post-stimulus time histograms (PSTHs) obtained from each network. There were strong time-locked responses in the early time window (<10 ms) from the stimulated networks (Fig. 6h, Elec stim $N_1$-Rec $N_1$, Elec stim $N_2$-Rec $N_2$, Elec stim $N_3$-Rec $N_3$). In the case of inter-network responses, the only response pair was $N_3$ (stimulation) and $N_2$ (recording). When $N_3$ was stimulated, evoked responses were detected only in $N_2$ between 15 and 35 ms after stimulation with a probability of 0.52, indicating that there was a functional connection from $N_3$ to $N_2$. The network mapping from spontaneous activity and electrically evoked responses confirmed that thermoplasmonic neural modulation can be used to analyze engineered neuronal networks in situ. Therefore, we have successfully demonstrated that the same thermoplasmonic interface employed for ablations can be utilized to estimate the functional connectivity of neuronal networks by adjusting the extent of thermoplasmonic heating.

## Discussion
In this study, we developed a neural chip platform using a GNR-assisted thermoplasmonic interface on an MEA to manipulate a cultured neuronal network both structurally and functionally in an in situ manner. The advantage of our platform is that it enabled three different in situ manipulations (Fig. 1b) by simply controlling the input power density of the NIR light source, while the development of electrical activity in neuronal networks was monitored non-invasively. Through the thermoplasmonic ablation of agarose hydrogel, new physical connections could be induced between matured networks, and the activity

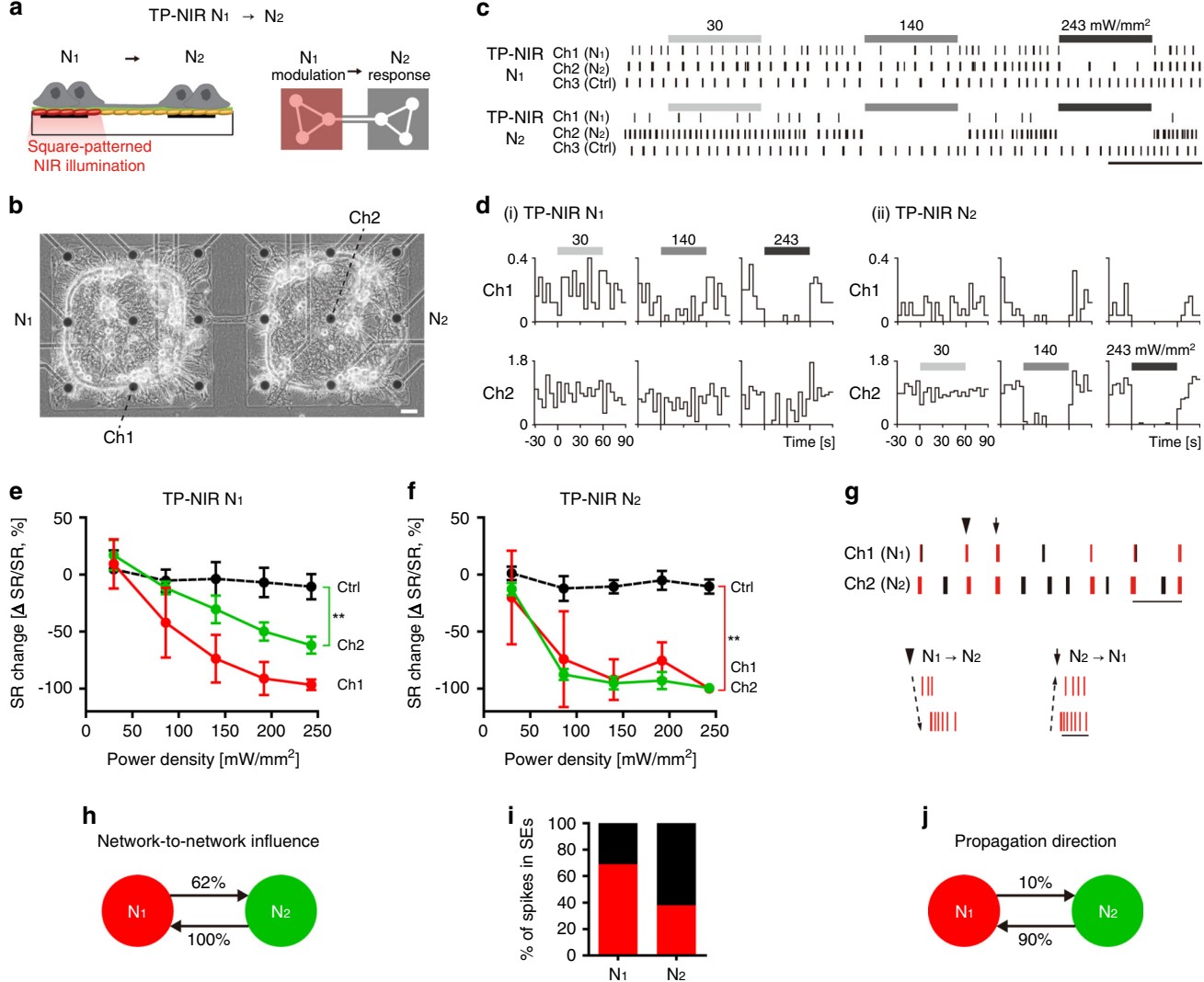

**Fig. 5 Thermoplasmonic neuromodulation and estimation of network connectivity. a** Thermoplasmonic NIR stimulation (TP-NIR) by square-patterned illumination. To estimate the network-to-network influence of $N_1$ on $N_2$ (TP-NIR $N_1 \rightarrow N_2$), $N_1$ spiking activity was suppressed by localized thermoplasmonic neural stimulation. **b** Phase-contrast image of tested network. Agarose micropatterns were 500 μm by 500 μm with a spacing of 100 μm. Scale bar: 50 μm. **c** Representative raster plots and (**d**) perievent histograms before, during, and after stimulation with different power densities (30, 140, 243 mW/mm²). TP-NIR $N_1$ and TP-NIR $N_2$ indicate the stimulation of $N_1$ and $N_2$, respectively. Scale bar: 1 min. x-axis: second. y-axis: Hz (spikes/sec). **e** Spike rate change against power density for TP-NIR $N_1$. The black line (Ctrl) shows rate change in control groups (5 electrodes in unconnected networks) in the same MEA. $n = 5$ trials for each network. Data are presented as mean ± SD. Two-sided Mann–Whitney test for the values of each network compared with those of control groups at the highest power density (**$p < 0.01$); $p = 0.0079$. **f** Spike rate change against power density for TP-NIR $N_2$. The black line (Ctrl) shows rate change in control groups (5 electrodes in unconnected networks) in the same MEA. $n = 5$ trials for each network. Data are presented as mean ± SD. Two-sided Mann–Whitney test for the values of each network compared with those of control groups at the highest power density (**$p < 0.01$); $p = 0.0075$. **g** Raster plots of spontaneous activity and examples of inter-network synchronized events (SEs) that spikes propagate from $N_1$ to $N_2$ (arrowhead) and from $N_2$ to $N_1$ (arrow). Red timestamps represent spikes that participated in SEs. Scale bar: 10 s and 500 ms. **h** Network-to-network influence estimated by thermoplasmonic neuromodulation. **i** Association degree in inter-network SEs. Red is the percentage of spikes participated in SEs. **j** Proportion of propagation direction within individual SEs. 106 detected SEs were analyzed. Source data are provided as a Source Data file.

synchronization of networks was assessed using the spontaneous activity recordings over several weeks (Fig. 3). The established functional connectivity was examined using more active intervention through the irreversible disconnection by neurite ablation (Fig. 4) and the reversible activity suppression by thermoplasmonic neuromodulation (Figs. 5, 6). Moreover, the agarose hydrogel cell patterning and spatially selective neurite guidance process greatly reduced the randomness of the global network architecture, which allowed us to precisely target the core connections to be removed or modulated. By the removal of physical connections or functional activity in network modules, the impact

of core connections in a modular network could be evaluated via electrical recordings.

The thermoplasmonic GNR-based interface in our platform provided several advantages. First, we were able to maximize the photothermal conversion efficiency by tuning the absorption peak of the GNRs at the desired wavelength. For this reason, we could ablate a conventional agarose hydrogel with a higher melting point (87–89 °C) using much lower laser power (from 20 mW) than that using a laser with the wavelength that is absorbed by water and hydrogel itself (melting temperature: 65 °C, wavelength: 1480 nm, laser power: 251 mW)[22]. Second, as our

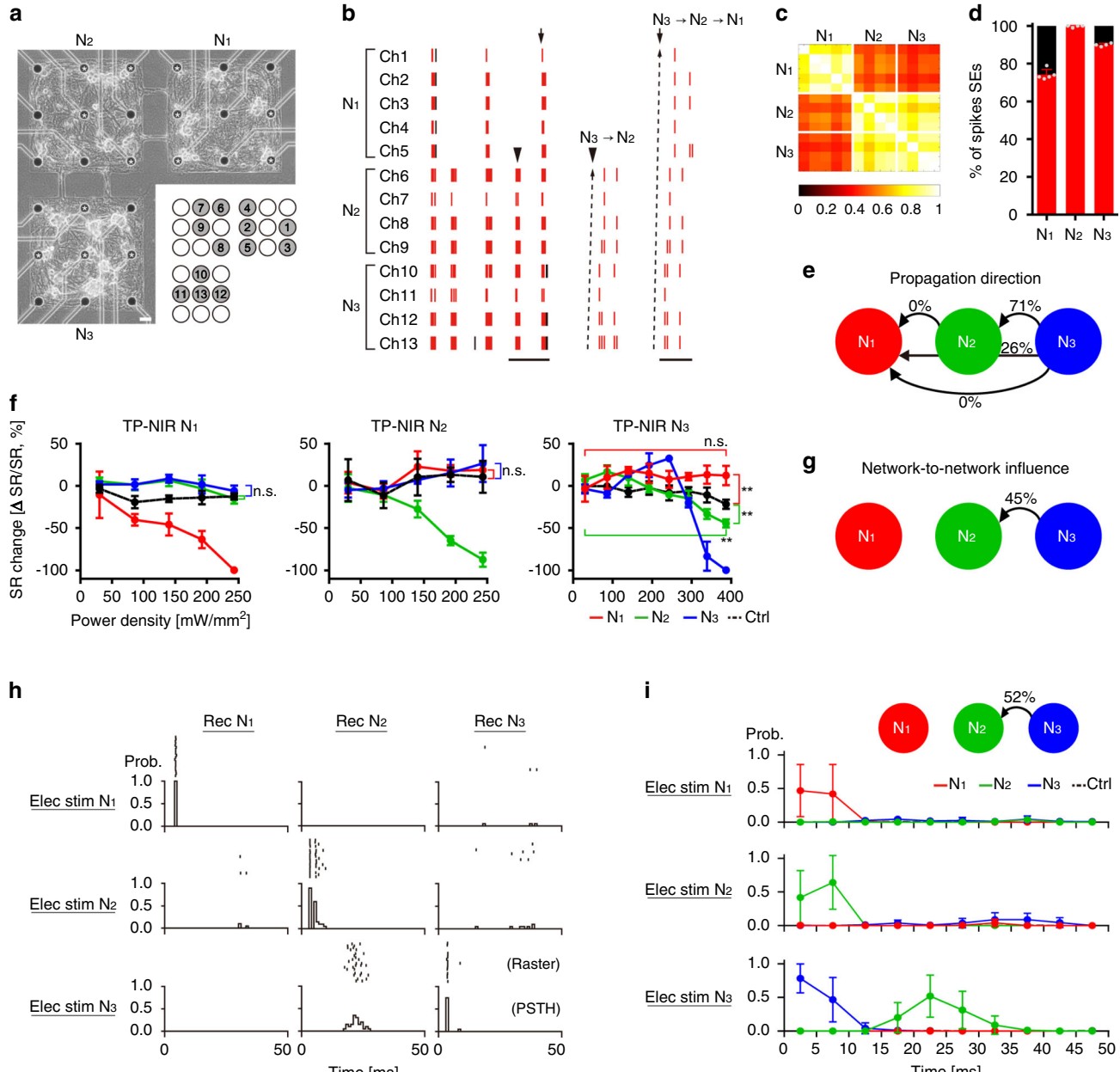

**Fig. 6 Thermoplasmonic neuromodulation of a three-module network. a** Phase-contrast image of tested network. Agarose micropatterns were 500 µm by 500 µm with a spacing of 100 µm. The asterisks indicate recording electrodes and the corresponding numbers are marked on an MEA layout. Scale bar: 50 µm. **b** Raster plots of spontaneous activity from 13 electrodes and examples of inter-network synchronized events (SEs) that spikes propagate from $N_3 \rightarrow N_2$ (arrowhead) or from $N_3 \rightarrow N_2 \rightarrow N_1$ (arrow). Red timestamps represent spikes that participated in SEs. Scale bar: 10 s (left) and 200 ms (right). **c** Correlation matrix at 38 DIV. **d** Association degree in SEs. Red is the percentage of spikes participated in SEs. $n = 5, 4, 4$ electrodes for $N_1, N_2, N_3$, respectively. Mean ± SD. **e** Proportion of propagation direction within individual SEs. 552 detected SEs were analyzed. **f** Spike rate changes against power density for TP-NIR $N_1$, TP-NIR $N_2$, and TP-NIR $N_3$. The black line (Ctrl) shows rate change in control groups (3 electrodes in unconnected networks) in the same MEA. $n = 5$ trials for each network. Mean ± SD. Two-sided Mann–Whitney test (**$p < 0.01$); comparison with control groups at the highest power density: $p = 0.8413, 0.0952$ ($N_2, N_3$) for TP-NIR $N_1$; $p = 0.4206, 0.3095$ ($N_1, N_3$) for TP-NIR $N_2$; $p = 0.0079, 0.0079$ ($N_1, N_2$) for TP-NIR $N_3$; comparison with the values at the lowest power density: $p = 0.1508, 0.0079$ ($N_1, N_2$) for TP-NIR $N_3$; **g** Network-to-network influence estimated by thermoplasmonic neuromodulation. **h** Representative PSTHs for electrical stimulation of a single electrode in each network. Bin size: 1 ms. **i** Mean PSTHs of all recording electrodes except stimulated ones. Stimulation electrodes: Ch3 and Ch5 in $N_1$, Ch6 and Ch9 in $N_2$, Ch12 and Ch13 in $N_3$. 20 trials for each electrode. Bin size: 5 ms. $n = 8, 8, 8$ electrodes for Elec stim $N_1$; 10, 6, 8 electrodes for Elec stim $N_2$; 10, 8, 6 electrodes for Elec stim $N_3$ ($N_1, N_2, N_3$, respectively); 10 electrodes for Ctrl. Mean ± SD. Source data are provided as a Source Data file.

platform was based on NIR wavelength (785 or 808 nm), the current technique was less harmful to cells compared to the method using UV light and much more localized without absorption of illumination light into the surrounding water and hydrogel (Supplementary Fig. 6)[19]. Third, the GNR interface was

stable and reliable under the cell culture environment. After thermoplasmonic hydrogel ablation, the GNRs on a substrate were well immobilized and their ability to generate local heat was not degraded, which allowed us to reapply thermoplasmonic stimulation for neurite ablation.

Our demonstrations have some implications in network physiology of in vitro modular neuronal networks. First, there was a significant time delay between the neurite touch-down and the emergence of SEs. There were several days (e.g., 2–6 days, Fig. 3f) of the time window between axonal extension to the target network and the onset of activity synchronization, which was considered as the time for target selection and synapse formation. Similar time delays have been reported in organotypic hippocampal slice cultures, where the burst activities of slices became synchronized 4–6 days after forming axonal connections[32]. Second, it was found that the functional integration between two matured networks, which had already formed intrinsic network activity, showed typical modular network behaviors. During the integration processes, there was an increase of globally SEs, while each network maintained its own activity (Fig. 3f). Such coexistence of globally integrated activity and locally segregated activity is known to be one of the representative features in modular neuronal networks[33]. The presence of segregated behavior could be clearly confirmed by the localized activity that occurred when the connections between the networks was removed (Fig. 4) or one of the network activities was suppressed (Figs. 5, 6). Third, the asymmetrical functional influence implied that there was a network driving module in the functionally integrated modular networks. The functional asymmetry has been reported in an in vitro network model that was designed to have a modular structure from the beginning of the cultivation[34,35]. We found that this intrinsic property was also present when several independent networks with strong intrinsic synchrony were used to construct a modular network. Since the proposed platform can investigate the network characteristics before connecting modules, further study will be able to reveal which innate factors are critical in determining the driving module.

From a technological point of view, the technique of in situ manipulation using our platform was capable of implementing an in vitro experimental model that mimics the modular architecture often found in the brain. Although most of the demonstrations presented in this work used two networks, the proposed platform can construct various sizes and numbers of modules with strong intra-connections, and can generate inter-modular connections between network modules at a desired time window and locations. Moreover, it would be possible to directly investigate the causal relationship between the induced physical connections and the emergence of new global activity by selectively removing the connections between modules or by reversibly suppressing the module activity. In addition to the inhibitory effect through thermoplasmonic stimulation, it is expected that advanced analysis can be performed by incorporating diverse modulations through electrical or chemical stimulation. Furthermore, combining our technique with the latest MEA technology will also be a powerful tool to enable mapping synaptic connectivity using intracellular recording[36].

## Methods

**Gold nanorod synthesis and immobilization on the culture substrates**. GNRs were synthesized through a seed-mediated approach introduced in a previous report[37]. To prepare the gold nanoseed as a template, 2.5 ml of 0.5 mM $HAuCl_4$ (Sigma-Aldrich, MO, USA), 2.5 ml of 0.2 M Cetyltrimethylammonium bromide (CTAB; Sigma-Aldrich), and 300 μl of 0.01 M $NaBH_4$ (Sigma-Aldrich) were mixed using an ultrasonic bath for 4 min at 25–28 °C. After aging of the nanoseeds, 12 μl of seed solution was added to a mixture of 5 ml of 1 mM $HAuCl_4$, 5 ml of 0.2 M CTAB, 280 μl of 4 mM $AgNO_3$ (Sigma-Aldrich), and 70 μl of 78.8 mM ascorbic acid (Sigma-Aldrich) for growing into rod-shaped structures. Following washing and collecting the GNRs using a centrifuge at 10000 rpm for 15 min, the obtained GNRs were reacted with thiol-polyethylene glycol-amine ($NH_2$-PEG-SH; Laysan Bio, AL, USA) for modifying their surface with PEG molecules. The synthesized GNRs showed a maximum absorption peak at 789 nm in water and their zeta potential was +26.5 mV. The GNRs were immobilized on a culture substrate, e.g., a glass coverslip or an MEA (60MEA200/30iR-ITO; Multichannel Systems,

Germany), as previously reported[31]. By activation of the substrates with air plasma (70 W, 1 min, Cute; Femto Science, South Korea), the amine-terminated PEG of GNRs was attached to the surface electrostatically. GNR solution with 1 optical density (OD) was dropped on the plasma-treated surface and kept in an incubator overnight.

**Fabrication of agarose hydrogel patterns on the GNR-coated substrate**. Before patterning the agarose hydrogel, the surface of GNR-coated substrates was incubated with 0.1 mg/ml poly-D-lysine (PDL; Sigma-Aldrich) in 10 mM Tris buffer at least 12 h to render it cell-adhesive. For damage characteristics of a biomolecule, 0.01 mg/ml of laminin (Mouse protein, Natural; Gibco; Thermo Fisher Scientific, MA, USA) in phosphate-buffered saline (PBS; Gibco) was loaded on the PDL layer for 1 h and washed with de-ionized (DI) water, followed by fully drying in a laminar flow hood.

For fabricating micropatterns of agarose hydrogel, polydimethylsiloxane (PDMS) molds were prepared by soft-lithography. A master for the PDMS mold was first fabricated by patterning a negative photoresist (SU-8 2010; MicroChem, MA, USA) of 10 μm height on a silicon wafer via a standard photolithography process. To facilitate the release of the PDMS mold, silane (Trichloro(1H, 1H, 2H, 2H-perfluorooctyl) silane; Sigma-Aldrich) was vapor-deposited on the SU-8 master in a vacuum desiccator. A mixture of PDMS prepolymer and curing agent (Sylgard 184; Dow Corning, MI, USA) in the ratio of 10: 1 was cast onto the master, degassed in the desiccator, and cured in a conventional oven at 60 °C for 3 h. After peeling off the PDMS mold, it was cut into pieces of the size of 4 mm by 4 mm square. Using a hardened PDMS mold as a template, agarose hydrogel was patterned on the PDL-coated substrates through a technique of micro-molding in capillaries developed in previous works[13,38]. The mold placed on a dummy glass was air plasma treated for 6 min to make it hydrophilic except for the surface to be in contact with the substrates. After aligning the plasma-treated mold on the substrate, agarose solution (2% w/v in DI water, melting temperature: 87–89 °C, Amresco Inc., OH, USA) was gently injected between the mold and substrate by capillary force. After filling with the hydrogel, the gelation and drying process was carried out in a 4 °C refrigerator for 30 min while sealing the substrates using parafilm and for 3 h without the sealing, respectively, followed by removing the PDMS mold from the substrate.

**Cell culture**. Hippocampal neurons were prepared by dissecting hippocampi from embryonic day 18 Sprague-Dawley rats (DBL, South Korea) and dissociating them in Hank's buffer salt solution (Welgene, South Korea) using a micropipette. The cell suspension was obtained by centrifuging for 2 min at 1000 rpm. After sterilizing the micropatterned substrates using 70% ethanol, the cells in a plating medium, which contained Neurobasal medium (Gibco), B-27 supplement (Gibco), 2 mM GlutaMAX (Gibco), 1% penicillin-streptomycin (Gibco), and 12.5 μM L-glutamate (Sigma-Aldrich), were seeded with the density of 600 cells/mm². The culture was kept in an incubator at 37 °C and 5% $CO_2$, and half of the medium was changed to a fresh maintenance medium (plating medium without L-glutamate) twice a week. All experiments were performed in accordance with the guidance of the Institutional Animal Care and Use Committee (IACUC) of Korea Advanced Institute of Science and Technology (KAIST), and all experimental protocols were approved by IACUC of KAIST.

**NIR illumination system**. Optical instrumentation for thermoplasmonic stimulation was modified from previous studies (Supplementary Fig. 1)[24,39]. For thermoplasmonic ablation of agarose hydrogel and neurites, a continuous NIR laser light (785 nm, 450 mW max, B&W Tek, DE, USA) was delivered through a collimator ($f = 8$ mm, NA = 0.5, Thorlabs, NJ, USA). By reflecting it on a dielectric mirror (750–1100 nm, Thorlabs) and a silver mirror (Thorlabs) sequentially, the collimated beam was fed into an inverted microscope (IX71, Olympus, Japan), reflected by a dichroic mirror (over 95% reflection at 785 nm, Chroma, VT, USA), and focused with an objective lens (50x, NA = 0.65, Olympus). The position of the focused beam whose diameter was about 20.5 μm was monitored using a CMOS camera (C11440; Hamamatsu, Japan) and the movement was controlled by a motorized stage (Ludl Electronic Products Ltd. NY, USA) with a custom-developed GUI program based on Microsoft Foundation Classes.

For a patterned illumination in thermoplasmonic neuromodulation, 808 nm laser (4W max, Laserlab, South Korea) was used as a light source. The light pattern was generated through a digital micromirror device (DMD; Texas Instruments, TX, USA) and focused on the substrate with an objective lens (4x, NA = 0.13, Olympus). The size of the illumination was 500 μm by 500 μm and all electrode areas were excluded by masking with a diameter of 41.25 μm to minimize electrical noise due to laser irradiation.

**Neural recording and connectivity analysis**. A custom-built amplifier (gain: 1000, band-pass filter: 150–4000 Hz) was used for signal recording. The signals were digitized at a sampling rate of 25 kHz by MC Card (Multichannel Systems) and recorded for 10–30 min using commercial software (MC_Rack; Multichannel Systems). Before the recording session, the culture was stabilized for at least 15 min by putting the MEA on a temperature controller (37 °C, TC01; Multichannel Systems) and feeding a 5% $CO_2$ gas. Neural spikes were extracted from the

recorded data by setting the threshold at −6 standard deviation of the background noise level.

To investigate the functional connectivity of neuronal networks, the correlation coefficient and array wide spike rate were obtained based on the spike timestamps from the electrodes whose firing rate was greater than 0.05 Hz (spikes/sec). The rate histogram with 100 ms bin was obtained for each electrode and smoothed with a Gaussian filter of five bins using NeuroExplorer (Nex Technologies, AL, USA). Pearson correlation coefficients for each pair of binned spike rates were calculated using MATLAB (MathWorks Inc., MA, USA) to construct a correlation matrix (function: corrcoef). To identify inter-network SEs, we used the detection method of synchronized bursting events (SBEs) in a previous study[40]. First, we detected bursts for each electrode by identifying successive spikes within a threshold window of 200 ms on individual electrodes. Then, SBEs were defined as the time windows in which bursts occur simultaneously in two or more electrodes, and SEs were defined as the SBEs in which two or more networks participate. For the analysis of Fig. 5, if there was at least one spike of Ch2 in the burst window of Ch1 or vice versa, this window was also considered as an SE. Association degree was calculated by dividing the number of spikes that participated in SEs by the total number of spikes in each electrode. Propagation direction of the SEs was determined based on the relative timing of the first spike at each electrode in individual SEs. All statistical data were plotted as mean and standard deviation (SD) and tested with Mann–Whitney test (**$p < 0.01$) using GraphPad Prism (GraphPad Software Inc., CA, USA).

**Electrical stimulation**. A positive first biphasic voltage pulse (amplitude: 500 mV, pulse width: 200 μs) was delivered to a single electrode using a stimulus generator (STG4004; Multichannel Systems). Twenty voltage stimuli were delivered every 3 s. Two electrodes in each network were tested and evoked spikes within 50 ms were counted. In the recording electrodes, stimulation artifact waveforms were excluded from regular spikes using a spike sorting technique (unit sorting with valley-seeking algorithm, OfflineSorter, Plexon Inc., TX, USA).

**Immunostaining and imaging**. To explore the neurite outgrowth, the immunostaining was performed as follows. Cultured neurons were fixed with 4% paraformaldehyde (Sigma-Aldrich) for 20 min at 4 °C and wash three times with PBS for 5 min each. To permeate the cell membrane, 1% Triton X-100 (Sigma-Aldrich) was used for 5 min at room temperature, followed by PBS washing. For blocking nonspecific binding, the sample was treated with 6% bovine serum albumin (BSA; Sigma-Aldrich) for 30 min. After washing with PBS, it was reacted with primary antibodies diluted in 1.5% BSA solution for 2 h at 37 °C. The following primary antibodies were used: anti-MAP2 (1:500, M3696, Sigma-Aldrich), anti-tau-1 (1:500, MAB3420, Merck Millipore, MA, USA), anti-beta-III-tubulin (1:500, T2200, Sigma-Aldrich). Secondary antibodies (Alexa Fluor 488 and 594, 1:500, A11001, A11008, A11012 and A11032, Invitrogen; Thermo Fisher Scientific) diluted in 1.5% BSA were loaded for 30 min at 37 °C. To characterize damage of a biomolecule on the NIR-illuminated surface, anti-laminin (1:500, L9393, Sigma-Aldrich) was used with the same procedure except that there is no cell fixation and permeabilization step. An inverted microscope (IX71; Olympus) with a digital camera (DP71; Olympus) was used to take fluorescence and phase-contrast images.

**Reporting summary**. Further information on research design is available in the Nature Research Reporting Summary linked to this article.

## Data availability

The data that support the findings of this study are available from the corresponding author upon reasonable request. Source data are provided with this paper.

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

## Acknowledgements

This work was supported by National Research Foundation grants (NRF-2015R1A2A1A09003605, NRF-2018R1A2A1A05022604, NRF-2016H1A2A1907681) funded by Korean government.

## Author contributions

N.H. and Y.N. conceived and designed the experiments. N.H. performed the experiments. N.H. and Y.N. analyzed the data and wrote the paper.

## Competing interests

The authors declare no competing interests.
