## [Peer Review File · Nature Communications]

Reviewers' Comments:

Reviewer #1:

Remarks to the Author:

I have read the manuscript with a great interest. The authors have an MEA platform with thermoplasmonic gold nanorod layers, and by using the latter for local heat modulation, they seek to modulate the neuronal connectivity more dynamically, i.e., even during the development and mature stages of the in vitro culture (as opposed to just setting it fixed once grown with a static substrate structure) and to see how the resulting electrophysiological signalings change (ultimately, how the neuronal network's function changes with the neuronal network's connectivity changed). The work is quite meaningful and can be very useful. It is also well written. I recommend the work to be published in Nature Communications, but after citing the following very relevant and timely article they have missed: Nature Biomedical Engineering 4, 232 (2020). As compared to the traditional MEA, the platform in the Nat. Biomed. Eng paper performs not only the extracellular recording of the network (as the MEA) for long-term monitoring, but also the intracellular recording of the network thus its synaptic signals (PSPs, for instance) as well as action potentials (advanced from the MEA) for the actual explicit synaptic connectivity mapping. So it would be actually quite meaningful to see, sometime in the future, how the heat modulation can actually modify the synaptic connectivity map and its strength matrix.

Reviewer #2:

Remarks to the Author:

The article by Hong et al. is an interesting and well-written article detailing three procedures to shape the morphology and influence the activity of neuronal networks. Specifically, the authors describe methods to (i) establish new connections between regions of a standard 64-electrode MEA that are populated with neurons and initially separated by an agarose gel layer, to (ii) the also cut the respective connections and (iii) to influence the activity of neurons in the compartments.

All this modulation is done via a gold-nanorod (GNR) layer deposited on top of the MEA and by directing a laser on selected areas of the GNR layer that then heats up to a varying degree to either just reduce activity of the neurons on top (iii), or to lead to a destruction of neuronal connections (ii) or to ablation of the agarose gel to enable neurite outgrowth (i).

While the paper is clearly written, and the strategy and findings are nicely explained. I am a bit concerned about (a) novelty/originality and (b) the low number of electrodes and neuronal signals that serve as evidence.

(a) While it is certainly the 1st time that such comprehensive use of all three procedures (new connections, remove connections, influence activity) is described and demonstrated with one platform, previous publications, also by the authors do already report on many of the components. The authors themselves have used the GNR layer for influencing the activity of neurons as reported in Refs. 24 and 32. The ablation of agarose gel, however through direct ablation of the gel by a laser, not via GNR layer heat-up has been amply used and described by the group of Jimbo (e.g., Refs. 21, 22) to shape and establish neuronal connections. Moreover, the cutting of neuronal connections by direct laser ablation (doi: 10.1117/1.3560268, doi: 10.3390/molecules21081018) has been used and described in vitro and by other authors also in vivo.

(b) It is not clear to me how many chips were used for the study, as there is no statistics no error bars in activities, only correlation diagrams, which seem to rely on measurements of activity of 2-3 electrodes, each of which may record from several neurons, as the electrodes are rather large and placed at a large distance.

In particular when the authors make claims about functional dependence between networks (lines

328 – 330, Figs. 3, 4, 5), such claims need to be substantiated with more measurements and several cultures or MEA dishes. There is no whatsoever information on how many cultures or dishes were used and what the sample-to-sample variation was and how successful the GNR method was in all those instances (Figs. 3, 4, 5).

A minor question that occurred to me is, why the neurons did not enter the ablated agarose channels, although those were wide enough upon using larger laser power. Or is it just a question of observation time and neurons migrated into those channels after a certain time?

Reviewer #3:

Remarks to the Author:

SYNOPSIS: The present manuscript describes the functional modifications of a microelectrode array such that connectivity within a neural network is induced by ablation of agarose hydrogel tracks or reduced by either ablating connectivity between neuronal networks or by partially inhibiting activity.

The approach described here is appealing. However, the authors need to better clarify the goal of the study. The title and the first two paragraphs of the discussion suggest that their goal is to demonstrate a new technological platform without demonstrating new biological findings (see line 312 "Our demonstrations showed biological findings comparable with those of previous studies"). The current manuscript describes three "biological findings" / in situ functional manipulations" (which are actually nicely presented in Fig.1B) . It provides little characterization of the chip platform, except for the brief description of Figure 1c-e and Figure 2a-b.

For instance, in the discussion section the authors claim:

line 301: "We were able to maximize the photothermal conversion efficiency by tuning an absorption peak of the GNRs at the desired wavelength." It is not clear how the authors "tuned" the peak. If this aspect is critical to the manuscript I would suggest to compare it with other GNRs.

Line 306: "current technique was less harmful to cells compared to the method using UV light and much localized without absorption of illumination light into the surrounding water and hydrogel (ref. 19)." It is difficult to judge if the technique presented is less harmful, given that in Fig.5 the authors present activity from one electrode out of nine in each compartment. Can the authors make sure that the MEA electrodes themselves or the activity on the remaining electrodes is not affected by the light.

Thirdly, NIR GNR are commercially available, i.e. from Nanopartz. Is there anything special about the GNR used here ?

In summary, although the chip platform may be novel there is too little evidence provided to support this claim.

From an application perspective, the induced network connectivity is interesting but shows only connectivity between neural networks not between individual cells. I would suggest focusing and exploring this part of the application. Why is there activity in compartment N2 not reflected in N1 ? How does the connectivity compare to the width of the ablated channel ?

The modulation of activity by NIR light is interesting but may be feasible with other modalities as well. For instance, one could infer network connectivity by other stimulation methods (optogenetics, electrical etc.). In principle, connectivity could be inferred by cross correlating the activities between compartments and accounting for the time lag between the correlation peak. The evidence provided in Fig.5 is very weak. It seems that only one channel in each compartment is considered and analyzed. How does the activity on other electrodes in compartment N1 and/or

compartment N2 look like ? The authors should carefully rethink or reformulate this theird example.

In summary, I find the novelty from the application perspective too limited in its current version. I encourage the authors, however, to address my concerns and provide a detailed roadmap & evidence how to build neural networks in situ using the approach provided i.e. in Figure 3.

Reply to referees' comments

< Reviewer: 1 >

I have read the manuscript with a great interest. The authors have an MEA platform with thermoplasmonic gold nanorod layers, and by using the latter for local heat modulation, they seek to modulate the neuronal connectivity more dynamically, i.e., even during the development and mature stages of the in vitro culture (as opposed to just setting it fixed once grown with a static substrate structure) and to see how the resulting electrophysiological signalings change (ultimately, how the neuronal network's function changes with the neuronal network's connectivity changed). The work is quite meaningful and can be very useful. It is also well written. I recommend the work to be published in Nature Communications, but after citing the following very relevant and timely article they have missed: Nature Biomedical Engineering 4, 232 (2020). As compared to the traditional MEA, the platform in the Nat. Biomed. Eng. paper performs not only the extracellular recording of the network (as the MEA) for long-term monitoring, but also the intracellular recording of the network thus its synaptic signals (PSPs, for instance) as well as action potentials (advanced from the MEA) for the actual explicit synaptic connectivity mapping. So it would be actually quite meaningful to see, sometime in the future, how the heat modulation can actually modify the synaptic connectivity map and its strength matrix.

>> In response to your comment, we have cited the article and added the following sentence in discussion:

“Furthermore, it will also be a powerful tool by combing our technique with the latest MEA technology that enable mapping synaptic connectivity using intracellular recording.³⁶”

(36) Abbott, J. et al. A nanoelectrode array for obtaining intracellular recordings from thousands of connected neurons. Nat. Biomed. Eng. (2020). doi:10.1038/s41551-019-0455-7

< Reviewer: 2 >

The article by Hong et al. is an interesting and well-written article detailing three procedures to shape the morphology and influence the activity of neuronal networks. Specifically, the authors describe methods to (i) establish new connections between regions of a standard 64-electrode MEA that are populated with neurons and initially separated by an agarose gel layer, to (ii) the also cut the respective connections and (iii) to influence the activity of neurons in the compartments.

All this modulation is done via a gold-nanorod (GNR) layer deposited on top of the MEA and by directing a laser on selected areas of the GNR layer that then heats up to a varying degree to either just reduce activity of the neurons on top (iii), or to lead to a destruction of neuronal connections (ii) or to ablation of the agarose gel to enable neurite outgrowth (i).

While the paper is clearly written, and the strategy and findings are nicely explained. I am a bit concerned about (a) novelty/originality and (b) the low number of electrodes and neuronal signals that serve as evidence.

(a) While it is certainly the 1st time that such comprehensive use of all three procedures (new connections, remove connections, influence activity) is described and demonstrated with one platform, previous publications, also by the authors do already report on many of the components. The authors themselves have used the GNR layer for influencing the activity of neurons as reported in Refs. 24 and 32. The ablation of agarose gel, however through direct ablation of the gel by a laser, not via GNR layer heat-up has been amply used and described by the group of Jimbo (e.g., Refs. 21, 22) to shape and establish neuronal connections. Moreover, the cutting of neuronal connections by direct laser ablation (doi: 10.1117/1.3560268, doi: 10.3390/molecules21081018) has been used and described in vitro and by other authors also in vivo.

>> As the reviewer pointed out, the novelty of this study is that all three manipulations are possible in one platform by simply controlling the input power density of NIR light source. We added the following paragraph in discussion to better highlight the advantage of our platform.

*“In this study, we developed a neural chip platform using a GNR-assisted thermoplasmonic interface on an MEA to manipulate a cultured neuronal network both structurally and functionally in an in situ manner. **The advantage of our platform was that it enabled three different in situ manipulations (Figure 1b) by simply controlling the input power density of NIR light source, while the development of electrical activity in neuronal networks were monitored non-invasively. Through the thermoplasmonic ablation of agarose hydrogel, new physical connections could be induced between matured networks, and the activity synchronization of networks were assessed using the spontaneous activity recordings over several weeks (Figure 3). The established functional connectivity was examined using more active intervention through the irreversible disconnection by neurite ablation (Figure 4) and the reversible activity suppression by thermoplasmonic neuromodulation (Figure 5).***

Moreover, the agarose hydrogel cell patterning and spatially selective neurite guidance process greatly reduced the randomness of a global network architecture structure, which allowed us to precisely target the core connections to be removed or modulated. By the removal of physical connections or functional activity in network modules, the impact of core connection of a modular network could be evaluated by electrical recordings.”

(b) It is not clear to me how many chips were used for the study, as there is no statistics no error bars in activities, only correlation diagrams, which seem to rely on measurements of activity of 2-3 electrodes, each of which may record from several neurons, as the electrodes are rather large and placed at a large distance.

In particular when the authors make claims about functional dependence between networks (lines 328 – 330, Figs. 3, 4, 5), such claims need to be substantiated with more measurements and several cultures or MEA dishes. There is no whatsoever information on how many cultures or dishes were used and what the sample-to-sample variation was and how successful the GNR method was in all those instances (Figs. 3, 4, 5).

>> We have added more measurements from several MEAs in main text and Supplementary information.

In case of neural bridging experiment (Figure 3e, 3f), we added more data set from 4 MEAs.

Figure 3. Network synchronization by creating new neurite connections. ... (e) Representative correlation matrices before and 3, 5, 7 days after ablation process. (f) Mean correlation coefficients of 4 MEAs. Green and blue graphs show inter-CC between networks with one and two-interconnection lines, respectively, and each line represents an individual MEA. An arrowhead indicates the values of the MEA in Figure 3e. Red graph and shaded region denote mean and SD values of intra-CC from 4 MEAs, respectively.

Figure S4. Network synchronization in large networks connected through various numbers of interconnection lines. ... (b) Correlation coefficient values within (intra-CC) and between (inter-CC) networks. Orange, green, blue, and purple graphs show inter-CC between connected networks with one, two, two, and three-interconnection lines, respectively. Each line represents an individual MEA. Intra-CC line and shaded region denote mean and standard deviation of intra-CCs from 4 MEAs, respectively.

Corresponding result section(‘Effect of new connections on network synchronization’) was revised:

“As shown in correlation matrices (Figure 3e), neurons formed local circuits in each network before the ablation, and spikes were only correlated between electrodes from same networks (N_1-N_1 , N_2-N_2 , N_3-N_3), while there was no correlation between networks (N_1-N_2 , N_1-N_3 , N_2-N_3). The connected networks (N_1-N_2) gradually became functionally synchronized between 3 and 7 days after the ablation (after 3 days: -0.002 , after 5 days: 0.273 , after 7 days: 0.460) and the correlation values reached close to those from local circuits (after 7 days; N_1-N_1 , N_2-N_2 : 0.660). Figure 3f shows the mean coefficient values from four different two-network samples that were connected after two weeks through one or two interconnection lines. The intra-correlation (intra-CC), which is the measure of the local synchrony in each module, had high coefficient values both before and after the ablation. This indicated that neurons in each network had established its own functional connections such that they tend to fire coincidentally even after new connections were formed with the other network (mean intra-CC; before: 0.613 , after 7 days: 0.775). In case of inter-correlation coefficients (inter-CC), which measures the functional synchrony of connected networks, the correlation values markedly increased after 2 to 6 days from the ablation. The increase of inter-CC reflected that neurons from different networks had tendency to participate in the globally synchronized events. The inter-CC from unconnected networks remained close to zero for seven days. Similar trends were also observed in larger networks connected through various numbers of interconnection lines (Figure S4). Taking together, we successfully induced structural changes by adding new neurite connections between the matured networks using thermoplasmonic ablation of agarose hydrogel, and traced the emergence of functional synchronization in connected networks over several weeks *in situ*. ”

In case of neural ablation experiment (Figure 4e, 4f), we added more data set from 3MEAs.

Figure 4. Thermoplasmonic neurite ablation and desynchronization between disconnected networks. ... (e) Representative correlation matrices before and after stimulation. (f) Mean correlation coefficients of 3 MEAs before (i) and after (ii) neurite ablation. Each graph indicates an individual MEA. Red graph shows the values of the MEA in Figure 4e.

Corresponding result section (‘Thermoplasmonic neural ablation and network desynchronization’) was revised:

“The array-wide rate histograms of two networks showed that the desynchronization between two networks occurred immediately after NIR illumination (Figure 4d). Furthermore, the network synchrony between two networks (N1-N2) dramatically declined to nearly zero (Figure 4e). As shown in Figure 4f, the inter-CC decreased in three different samples (from 0.310 to -0.018; from 0.499 to 0.015; from 0.519 to 0.016), while the intra-CC remained high (from 0.673 to 0.592; from 0.882 to 0.836; from 0.929 to 0.921). This indicated that individual networks still maintained the synchronized activities within themselves despite the removal of overall synchrony between networks. Thus, we concluded that the thermoplasmonic neurite ablation was sufficient to eliminate functional connectivity between synchronized networks, and this method could be used to modulate structural changes to study functional connectivity in neuronal networks.”

In case of thermoplasmonic neuromodulation test (Figure 6), we added another experiment set with more recording electrode. The corresponding section (‘Network mapping by thermoplasmonic neuromodulation’) was extensively revised by adding new connectivity analyses (functional connectivity analysis from spontaneous activity (Figure 6b-e) and electrical stimulation (Figure 6h,i)).

Figure 6. Thermoplasmonic neuromodulation of a three-module network. (a) Phase-contrast image of tested networks. Agarose micropatterns were 500 μm by 500 μm with spacing of 100 μm . The asterisks indicate recording electrodes. Scale bar: 50 μm . (b) Raster plots of spontaneous activity from thirteen electrodes and examples of inter-network SEs that spikes propagate from $N_3 \rightarrow N_2$ (arrowhead) or from $N_3 \rightarrow N_2 \rightarrow N_1$ (arrow). Red timestamps represent spikes participated in SEs. Scale bar: 10 sec and 200 ms. (c) Correlation matrix at 38 DIV. (d) Association degree in inter-network SEs. Red is the percentage of spikes participated in SEs. (e) Proportion of propagation direction within individual SEs. 552 detected SEs were analyzed. (f) Spike rate change against power density for TP-NIR N_1 , TP-NIR N_2 , and TP-NIR N_3 . Black line (Ctrl) shows rate change in control groups (3 electrodes in unconnected networks) in the same MEA. $n = 5$ trials for each network. (g) Network-to-network influence estimated by thermoplasmonic neuromodulation. (h) Representative PSTHs for an electrical stimulation of a single electrode in each network. Bin size: 1 ms. (i) Mean PSTHs of all recording electrodes except stimulated ones. Stimulation electrodes: Ch3 and Ch5 in N_1 , Ch6 and Ch9 in N_2 , Ch12 and Ch13 in N_3 . 20 trials for each electrode. Bin size: 5 ms.

A minor question that occurred to me is, why the neurons did not enter the ablated agarose channels, although those were wide enough upon using larger laser power. Or is it just a question of observation time and neurons migrated into those channels after a certain time?

>> When we performed the thermoplasmonic ablation 1 – 3 weeks after cell seeding, there were few cells that entered into ablated channels for 2 weeks. After establishing a robust network in the patterned agarose well,

the soma did not seem to move dramatically. According to our observation, the cells that actively migrated into the channels were progenitor cells rather than mature neurons because they were relatively small in size, and NeuN-negative and Nestin-positive.

< Reviewer: 3 >

SYNOPSIS: The present manuscript describes the functional modifications of a microelectrode array such that connectivity within a neural network is induced by ablation of agarose hydrogel tracks or reduced by either ablating connectivity between neuronal networks or by partially inhibiting activity.

The approach described here is appealing. However, the authors need to better clarify the goal of the study. The title and the first two paragraph of the discussion suggest that their goal is to demonstrate a new technological platform without demonstrating new biological findings (see line 312 “ Our demonstrations showed biological findings comparable with those of previous studies “).

>> To clarify the goal and the scope of the study, the introduction and discussion sections have been revised.

In introduction section,

“The goal of this study is to develop a technological platform for novel in situ manipulation techniques to study the structure and function of neural networks. This platform should be able to induce structural changes at a desired time point during neuronal development and maturation and investigate network connectivity affected by those changes.”

In result section ‘Effect of new connections on network synchronization’

“To generate new neurite connections between matured neuronal networks and investigate its impact on functional change in network connectivity, we designed a modular network using two unconnected networks (‘modules’) on an MEA and applied thermoplasmonic ablation to establish biological interconnections between two networks. As shown in Figure 3a, each agarose well pattern contained four microelectrodes and neurons within the well constituted an individual network. ... ”

In discussion section,

“Our demonstrations have some implications in network physiology of in vitro modular neuronal networks. First, there was a significant time delay between the neurite touch-down and the emergence of synchronized events. There were several days (e.g. 2 – 6 days, Figure 3f) of time window between axonal extension to the target network and the onset of activity synchronization, which was considered as the time for target selection and synapse formation. Similar time delays have been reported in organotypic hippocampal slice cultures, where the burst activities of slices became synchronized 4 – 6 days after forming axonal connections.³² Second, it was found that the functional integration between two matured networks, which had already formed intrinsic network activity, showed typical modular network behaviors. During the integration processes, there were the increase of globally synchronized events, while each network maintained their own activity (Figure 3f). Such coexistence of globally integrated activity and locally segregated activity is known to be one of the representative features in modular neuronal networks³³. The presence of segregated behavior could be clearly confirmed by the localized

activity that occurred when the connections between the networks was removed (Figure 4) or one of the network activities was suppressed (Figure 5 and 6). Third, the asymmetrical functional influence implied that there was a network driving module in the functionally integrated modular networks. The functional asymmetry has been reported in an in vitro network model that was designed to have a modular structure from the beginning of the cultivation.^{34,35} We found that this intrinsic property was also present when several independent networks with strong intrinsic synchrony were used to construct a modular network. Since the proposed platform can investigate the network characteristics before connecting modules, further study will be able to reveal which innate factors are critical in determining the driving module.”

The current manuscript describes three biological findings / in situ functional manipulations (which are actually nicely presented in Fig.1B). It provides little characterization of the chip platform, except for the brief description of Figure 1c-e and Figure 2a-b.

For instance, in the discussion section the authors claim:

Line 301: “We were able to maximize the photothermal conversion efficiency by tuning an absorption peak of the GNRs at the desired wavelength.” It is not clear how the authors “tuned” the peak. If this aspect is critical to the manuscript I would suggest to compare it with other GNRs.

>> It is well-known that a seed-mediated growth methods can tune a longitudinal plasmon resonance peak of gold nanorods by adjusting the silver ion content.¹ In this study, we tried to synthesize the GNRs with a longitudinal absorption peak (789 nm) close to the wavelength of the laser we used (mainly 785 nm).

In response to the reviewer’s comment, we have appended a data set that compared GNRs with shorter wavelength of longitudinal peaks (759 nm) in supplementary figure (Figure S3) and added the following sentence in main text:

“As we tuned the longitudinal absorption peak of GNRs (789 nm) close to the wavelength of the laser that was applied (785 nm), we could achieve high photothermal conversion efficiency for agarose ablation (Figure S3).”

(1) X. Huang, S. Neretina, M. A. El-Sayed, Gold nanorods: From synthesis and properties to biological and biomedical applications. *Adv. Mater.* **2009**.

Figure S3. Comparison of two different GNRs for the thermoplasmonic ablation of agarose hydrogel. (a) Absorbance spectra of GNRs with a longitudinal peak of 789 nm and 759 nm. (b) Phase-contrast images of ablated regions. Scale bar: 50 μm . (c) Ablated width of agarose hydrogel against power density for two different GNRs. The ablated width when using GNRs with 789 nm-peak was larger than that when using GNRs with 759 nm-peak. $n = 9$ (789 nm) and 4 (759 nm) ablated lines. mean \pm standard deviation.

Line 306: "current technique was less harmful to cells compared to the method using UV light and much localized without absorption of illumination light into the surrounding water and hydrogel (ref. 19)." It is difficult to judge if the technique presented is less harmful, given that in Fig.5 the authors present activity from one electrode out of nice in each compartment. Can the authors make sure that the MEA electrodes themselves or the activity on the remaining electrodes is not affected by the light.

>> Thank you for the comment. We have confirmed that there was no effect on the remaining electrodes by the light. We added a data set that shows the effect of the NIR laser on the remaining electrodes by comparing the noise level of electrodes (illuminated vs. non-illuminated) in supplementary information (Figure S6).

For five illumination trials with the power density of 243 mW/mm^2 , the normalized root mean square (RMS) values of background noise for electrodes inside and outside the NIR-illuminated region after each trial were calculated. In case of electrodes in NIR-illuminated region, there was no section showing a significant difference between trials (One-way ANOVA; $p = 0.8365$; $n = 18$ electrodes). After five trials of NIR illumination, the change in noise values of electrodes within NIR-illuminated region was not significantly different from that of controls that electrodes were outside the NIR-illuminated region, implying that the NIR light did not cause any significant damage to electrodes themselves (Unpaired t-test; $p = 0.9453$)

Figure S6. Comparison of background noise level after NIR illumination. (a) Change of RMS (root-mean squared) noise values after individual trials normalized by the RMS value before the illumination (243 mW/mm²). Red and black lines represent the values of electrodes inside (NIR+; n = 18 electrodes) and outside (NIR-; n = 36 electrodes) the NIR-illuminated region, respectively. In case of electrodes in NIR-illuminated region, there was no section showing a significant difference between trials (One-way ANOVA; p = 0.8365). (b) Change of RMS noise values after five illumination trials relative to those before the illumination. The change of NIR-illuminated region (NIR+) was not significantly different from that of controls (NIR-), implying that the NIR did not cause any significant damage to electrode themselves (Unpaired t-test; p = 0.9453).

Thirdly, NIR GNR are commercially available, i.e. from Nanopartz. Is there anything special about the GNR used here?

>> We prefer to synthesize our own GNRs due to the quality control, shelf life time, price, and reproducibility. Commercialized GNRs can be also used.

In summary, although the chip platform may be novel there is too little evidence provided to support this claim.

>> As the reviewer pointed out, we have added more data set to support our claim. Four data sets were added in the neural bridging experiment (Figure 3e, 3f), and three data sets were added in the neural ablation experiment (Figure 4e, 4f). To show the application of building neuronal networks with larger size, we added 3-network experiment results (Figure 6) including more recording electrodes and electrical stimulations.

Figure 3. Network synchronization by creating new neurite connections. ... (e) Representative correlation matrices before and 3, 5, 7 days after ablation process. (f) Mean correlation coefficients of 4 MEAs. Green and blue graphs show inter-CC between networks with one and two-interconnection lines, respectively, and each line represents an individual MEA. An arrowhead indicates the values of the MEA in Figure 3e. Red graph and shaded region denote mean and SD values of intra-CC from 4 MEAs, respectively.

Figure 4. Thermoplasmonic neurite ablation and desynchronization between disconnected networks. ... (e) Representative correlation matrices before and after stimulation. (f) Mean correlation coefficients of 3 MEAs before (i) and after (ii) neurite ablation. Each graph indicates an individual MEA. Red graph shows the values of the MEA in Figure 4e.

Figure 6. Thermoplasmonic neuromodulation of a three-module network. (a) Phase-contrast image of tested networks. Agarose micropatterns were 500 μm by 500 μm with spacing of 100 μm . The asterisks indicate recording electrodes. Scale bar: 50 μm . (b) Raster plots of spontaneous activity from thirteen electrodes and examples of inter-network SEs that spikes propagate from $N_3 \rightarrow N_2$ (arrowhead) or from $N_3 \rightarrow N_2 \rightarrow N_1$ (arrow). Red timestamps represent spikes participated in SEs. Scale bar: 10 sec and 200 ms. (c) Correlation matrix at 38 DIV. (d) Association degree in inter-network SEs. Red is the percentage of spikes participated in SEs. (e) Proportion of propagation direction within individual SEs. 552 detected SEs were analyzed. (f) Spike rate change against power density for TP-NIR N_1 , TP-NIR N_2 , and TP-NIR N_3 . Black line (Ctrl) shows rate change in control groups (3 electrodes in unconnected networks) in the same MEA. $n = 5$ trials for each network. (g) Network-to-network influence estimated by thermoplasmonic neuromodulation. (h) Representative PSTHs for an electrical stimulation of a single electrode in each network. Bin size: 1 ms. (i) Mean PSTHs of all recording electrodes except stimulated ones. Stimulation electrodes: Ch3 and Ch5 in N_1 , Ch6 and Ch9 in N_2 , Ch12 and Ch13 in N_3 . 20 trials for each electrode. Bin size: 5 ms.

To emphasize the novelty of our platform, we have added following paragraph in discussion.

“In this study, we developed a neural chip platform using a GNR-assisted thermoplasmonic interface on an MEA to manipulate a cultured neuronal network both structurally and functionally in an in situ manner. **The advantage of our platform was that it enabled three different in situ manipulations (Figure 1b) by simply controlling the input power density of NIR light source, while the development of electrical activity in neuronal networks were monitored non-invasively. Through the thermoplasmonic ablation of agarose hydrogel, new physical connections could be induced between matured networks, and the activity synchronization of networks were assessed using the spontaneous activity recordings over several weeks (Figure 3). The established functional connectivity was examined using more active intervention through the irreversible disconnection by neurite ablation (Figure 4) and the reversible activity suppression by thermoplasmonic neuromodulation (Figure 5). Moreover, the agarose hydrogel cell patterning and spatially selective neurite guidance process greatly reduced the randomness of a global network architecture structure, which allowed us to precisely target the core connections to be removed or modulated. By the removal of physical connections or functional activity in network modules, the impact of core connection of a modular network could be evaluated by electrical recordings.**”

From an application perspective, the induced network connectivity is interesting but shows only connectivity between neural networks not between individual cells. I would suggest focusing and exploring this part of the application.

>> We agree with the reviewer’s suggestion. We believe the network-to-network connectivity has been unexplored in synchronized modular networks and designed our experiments to examine the connectivity between networks. This was why we illuminated the NIR laser that covered the entire network (500 μm by 500 μm as shown in Figure 5a) rather than single cell or electrode to modulate the network-scale activity.

In our network analysis, we analyzed cell-to-cell connectivity to evaluate the network-to-network connectivity. In case of inducing or removing network connections (Figure 3, 4), correlation coefficients (intra-CC, inter-CC) have been used (Figure 3e – f, 4e – f).

In this revised manuscript, we added cell-to-cell connectivity analysis (spiking firing order in synchronized events, Figure 5g – j, 6b – e) in thermoplasmonic modulation experiments to compare our network-to-network connectivity measurement with cell-to-cell connectivity measure.

Why is there activity in compartment N2 not reflected in N1?

>> In the result of Figure 5, the activities of both compartments were reflected in the other compartment, but the degree was different (Influence from N1 to N2 was 62% and that from N2 to N2 was 100%). By analyzing the result of this thermoplasmonic neuromodulation with the spontaneous activity in the networks (Figure 5e – j), we found out that this asymmetric influence was related to the degree to which the spikes in the network participated with inter-network synchronized events and the order in which spikes propagated within each synchronized event.

We have explained the detail in the result section:

“The network-to-network influence was found to be unbalanced (Figure 5h, $I_{N1 \rightarrow N2}$: 62%; $I_{N2 \rightarrow N1}$: 100%). In other words, there was an asymmetrical dependency between two networks. Spiking activity of N_1 was 100% driven by external connections from N_2 , while some portion of the activity in N_2 (62%) was driven by external connections and the remaining portion was driven by the local recurrent circuits in N_2 . It implies that functional connectivity was established in both directions, but the degree of connectivity was stronger in $N_2 \rightarrow N_1$. To confirm our interpretation, we analyzed spatiotemporal patterns in spontaneous activity recordings. As shown in Figure 5g, both network modules had bursting activity patterns. Among these bursts, some of them were associated with inter-network synchronized events (SEs) and others were unassociated intra-network events. The percentage of total spikes participating in inter-network SEs were 69% and 38% for N_1 and N_2 , respectively (Figure 5i), which indicated that there was large difference in the degrees of association with the global network activity for two networks. Furthermore, the spike propagation within the inter-network SEs were dominantly from N_2 to N_1 (Figure 5j). In this synchronized network, network mapping using network-to-network influence measure was consistent with functional analysis of spontaneous activity.”

How does the connectivity compare to the width of the ablated channel?

>> As shown in Figure 2a, if the width is increased by illuminating the higher power density, the biomolecule can be damaged. Therefore, we expanded the width of interconnection channels by enlarging the number of channels with fixed power density. In Figure 3f and S4, it seemed that the time points at which the inter-correlation increased appeared to be faster as the number of connected channels between networks increased. Although this issue could not be concluded in this study, it is one of the topics that we would like to explore in future experiments.

The modulation of activity by NIR light is interesting but may be feasible with other modalities as well. For instance, one could infer network connectivity by other stimulation methods (optogenetics, electrical etc.). In principle, connectivity could be inferred by cross correlating the activities between compartments and accounting for the time lag between the correlation peak. The evidence provided in Fig.5 is very weak. It seems that only one channels in each compartment is considered and

analyzed. How does the activity on other electrodes in compartment N1 and/or compartment N2 look like? The authors should carefully rethink or reformulate this third example.

>> Thank you for the constructive comments.

First, concerning the other modalities to infer network connectivity, the advantage of using NIR light / GNR system over electrical stimulation or optogenetics are addressed. It is very hard to achieve reversible activity suppression with extracellular electrical stimulation. One can use optogenetics, but it needs genetic modification which might not be the best option for studying the developing neuronal networks.

We added the following sentence in Introduction:

“Finally, we show that the network connectivity of engineered modular networks can be investigated by the proposed thermoplasmonic platform at lower input power density levels. Based on our successful manipulation of network connections and observation of emergent synchronization, we attempted to estimate functional influence or dependency between network modules using our platform. According to previous studies, it is possible to suppress neural activity by delivering heat to neurons using thermoplasmonic gold nanoparticles.³¹ **Unlike electrical stimulation or optogenetics, thermoplasmonic neural stimulation can implement reversible neural suppression of the desired scale by controlling the light power and does not require genetic modification, which would be beneficial for mapping network connectivity in situ.**”

Second, as the reviewer pointed out, Figure 5 had only one electrode from each network. Unfortunately, there was only single activity electrode in each compartment on the day of neuromodulation test (19d after the ablation), which was mainly due to the cell movement. As shown in the figures below, the neurons in agarose patterns moved away from the edge to the center and clustered each other. At that same time, the number of electrodes exhibiting active neuronal spikes (white asterisk) decreased, especially as the electrodes close to the edge of the patterns failed to measure the spikes.

< Phase contrast micrograph of an MEA chip used in Figure 5 (19d after).
 N1: top compartment, N2: bottom compartment >

However, we can provide the spike trains from multiple electrodes in each compartment as we were able to track the activity from active electrodes (channels) after the ablation. The spike train clearly shows the synchronized activity, and correlation coefficients in each compartment (intra-CC) in the correlation matrices very high (0.7 ~ 0.9). Thus, we were certain that one channels in each compartment could represent each compartment activity to evaluate the interaction between two compartments at 19d after the ablation (Figure 5). To complement this data set, we have added another example with multiple active electrodes in Figure 6.

< (Top row) Spike trains from multiple active channels in each compartment (N1, N2) at 9d after and 11d after the ablation. (Bottom row) Correlation matrices evaluated at 9d and 11d after the ablation. An MEA chip in Figure 5. >

Third, to reformulate the third example section (‘Network mapping by thermoplasmonic neuromodulation’), the whole section has been rewritten with additional data analysis and experimental data set. The network mapping has been re-termed as ‘network-to-network influence’ instead of functional dependency to clarify the goal of demonstration. To validate the estimated network mapping, we presented the connectivity analysis from spontaneous activity and electrical stimulation, and compared with the thermoplasmonic connectivity mapping results (Figure 5g – j, Figure 6b – e, h, i).

The following is the revised section:

“Finally, we show that the network connectivity of engineered **modular** networks can be investigated by the proposed thermoplasmonic platform at lower input power density levels. **Based on our successful manipulation of network connections and observation of emergent synchronization**, we attempted to estimate **functional influence or dependency between network modules using our platform**. According to previous studies, it is possible to suppress neural activity by delivering heat to neurons using thermoplasmonic gold nanoparticles.³¹ Unlike electrical stimulation or optogenetics, thermoplasmonic neural stimulation can implement reversible neural suppression of the desired scale by controlling the light power and does not require genetic modification,

which would be beneficial for mapping network connectivity *in situ*. To measure the functional influence between network modules, the spiking activity in one module ('network N₁') was completely suppressed by thermoplasmonic NIR stimulation (TP-NIR) and the corresponding change from another network ('network N₂') was evaluated (Figure 5a). The network-to-network influence of network N₁ on N₂ ($I_{N_1 \rightarrow N_2}$) was defined as the baseline activity change in the network N₂ at the maximum suppression of the modulating network N₁.

First example shows a synchronized modular network composed of two networks (Figure 5b). Two networks in agarose micropatterns were connected with one interconnection line by ablating the hydrogel at 15 DIV, and they became synchronized from 22 DIV (Figure S4). The functional influence test was executed at 34 DIV when the synchrony measure was relatively high (inter-CC: 0.407). To eliminate the network activity completely, the NIR illumination pattern was designed to cover the whole area of one network module. The laser power densities for the modulation was much lower than those used for the agarose ablations (30 – 243 mW/mm² vs. 68 – 150 W/mm²). As shown in representative raster plots and perievent histograms (Figure 5c and d), modulating one network induced activity changes in the other network. When N₁ was fully suppressed (TP-NIR N₁, Ch1, 243 mW/mm²), the spiking activity in N₂ (Ch2) seemed to be perturbed. When N₂ was suppressed (TP-NIR N₂, Ch2, 140 mW/mm²), the spiking activity in N₁ (Ch1) activity was completely vanished, which implies that the N₁ spiking activity was highly influenced by N₂. Figure 5e and f show the spike rate change following the thermoplasmonic modulation in each network. Under the full inhibition of the modulated network, there were significant baseline activity changes for both cases (TP-NIR N₁ and TP-NIR N₂) compared to the spike changes in the control group modules that were not connected. When N₁ was modulated from 0 to – 97% (Figure 5e), N₂ activity changed from 0 to – 62%. When N₂ was modulated from 0 to – 99% (Figure 5f), N₁ activity changed accordingly (0 to – 100%). The network-to-network influence was found to be unbalanced (Figure 5h, $I_{N_1 \rightarrow N_2}$: 62%; $I_{N_2 \rightarrow N_1}$: 100%). In other words, there was an asymmetrical dependency between two networks. Spiking activity of N₁ was 100% driven by external connections from N₂, while some portion of the activity in N₂ (62%) was driven by external connections and the remaining portion was driven by the local recurrent circuits in N₂. It implies that functional connectivity was established in both directions, but the degree of connectivity was stronger in N₂ → N₁. To confirm our interpretation, we analyzed spatiotemporal patterns in spontaneous activity recordings. As shown in Figure 5g, both network modules had bursting activity patterns. Among these bursts, some of them were associated with inter-network synchronized events (SEs) and others were unassociated intra-network events. The percentage of total spikes participating in inter-network SEs were 69% and 38% for N₁ and N₂, respectively (Figure 5i), which indicated that there was large difference in the degrees of association with the global network activity for two networks. Furthermore, the spike propagation within the inter-network SEs were dominantly from N₂ to N₁ (Figure 5j). In this synchronized network, network mapping using network-to-network influence measure was consistent with functional analysis of spontaneous activity.

Next, we extended our analysis to a three-module network which were strongly synchronized. Three modular networks were connected with two interconnection lines at 21 days (Figure 6a). Seventeen days after connecting three modules (38 DIV), the network exhibited strong synchronized bursting patterns (Figure 6b). In each module, spikes from multiple electrodes showed high degree of synchrony (Figure 6c, mean intra-CC N1: 0.839, N2: 0.851, N3: 0.858). Among three modules, inter-network synchrony measures showed that they were highly synchronized among each other (Figure 6c and S5, mean inter-CC: 0.576, N1-N2: 0.513, N1-N3: 0.455, N2-N3: 0.806), and nearly all bursts were highly associated with inter-network SEs. (Figure 6d, N1: 74.2%, N2: 99.8%, N3: 90.0%). Among the SEs, two persistent activity patterns dominated the network-wide activity: N3 → N2 (71%) and N3 → N2 → N1 (26%) (Figure 6e). By thermoplasmonic modulation (Figure 6f), nearly all of the spikes in the modulated network were suppressed at the highest power density. Network-to-network influence measure revealed that there was only one significant connection pair in this network (Figure 6g, $I_{N_3 \rightarrow N_2}$:

45%), which corresponded to the persistent activity patterns found from spontaneous activity (Figure 6e, N3 → N2). We further analyzed the network connectivity using electrical stimulation and compared with our influence measure. Figure 6h and i show post-stimulus time histograms (PSTHs) obtained from each network. There were strong time-locked responses in early time window (< 10 ms) from the stimulated networks (Figure 6 h, Elec stim N1-Rec N1, Elec stim N2-Rec N2, Elec stim N3-Rec N3). In case of inter-network responses, the only response pair was N3 (stimulation) and N2 (recording). When N3 was stimulated, evoked responses were detected only in N2 between 15 and 35 ms after stimulation with the probability of 0.52, indicating that there was a functional connection from N3 to N2. The network mapping from spontaneous activity and electrically evoked responses confirmed that thermoplasmonic neural modulation can be used to analyze engineered cultured neuronal networks in situ. Therefore, we have successfully demonstrated that the same thermoplasmonic interface employed for ablations can be utilized to estimate the functional connectivity of neuronal networks by adjusting the extent of thermoplasmonic heating.”

As we added the analysis, the following method sections were revised:

Neural recording and connectivity analysis

“To investigate the functional connectivity of neuronal networks, the correlation coefficient and array wide spike rate were obtained based on the spike time stamps from the electrodes whose firing rate were greater than 0.05 Hz (spikes/sec). The rate histogram with 100 ms bin was obtained for each electrode and smoothed with a Gaussian filter of 5 bins using NeuroExplorer (Nex Technologies, AL, USA) and Pearson correlation coefficient for each pair of binned spike rate was calculated using MATLAB (MathWorks Inc., MA, USA) to construct a correlation matrix (function: corrcoef). To identify inter-network synchronized events (SEs), we used the detection method of synchronized bursting events (SBEs) in a previous study.⁴¹ We used a threshold value of 200 ms to detect bursts in individual electrodes. SBEs in which two or more networks participate was defined as SEs. For the analysis of Figure 5, if there was at least one spike of Ch2 in the burst window of Ch1, or vice versa, this window was also considered as a SE. Association degree was calculated by dividing the number of spikes participated in SEs by the total number of spikes in each electrode. Propagation direction of the SE was determined based on the relative timing of the first spike at each electrode in individual SEs. All statistical data was plotted as mean and standard deviation (SD) and tested with Mann-Whitney test (**p < 0.01) using GraphPad Prism (GraphPad Software Inc., CA, USA).

Electrical stimulation

A positive first biphasic voltage pulse (amplitude: 500 mV, pulse width: 200 μs) was delivered to a single electrode using a stimulus generator (STG4004; Multichannel Systems). Twenty voltage stimuli were delivered every 3 seconds. Two electrodes in each network were tested and evoked spikes within 50 ms were counted. In the recording electrodes, stimulation artifacts waveforms were excluded from regular spikes using a spike sorting technique (unit sorting with valley-seeking algorithm, OfflineSorter, Plexon Inc., TX, USA).”

Figure 5. Thermoplasmonic neuromodulation and estimation of network connectivity. (a) Thermoplasmonic NIR stimulation (TP-NIR) by square-patterned illumination. To estimate the network-to-network influence of N_1 on N_2 (TP-NIR $N_1 \rightarrow N_2$), N_1 spiking activity was suppressed by localized thermoplasmonic neural stimulation. (b) Phase-contrast image of tested networks. Agarose micropatterns were $500 \mu m$ by $500 \mu m$ with spacing of $100 \mu m$. Scale bar: $50 \mu m$. (c) Representative raster plots and (d) peri-event histograms before, during, and after stimulation with different power densities (30, 140, 243 mW/mm^2). TP-NIR N_1 and TP-NIR N_2 indicate stimulation of N_1 and N_2 , respectively. Scale bar: 1 min. x-axis: second. y-axis: Hz (spikes/sec). (e) Spike rate change against power density for TP-NIR N_1 . (f) Spike rate change against power density for TP-NIR N_2 . Black line (Ctrl) shows rate change in control groups (5 electrodes in unconnected networks) in the same MEA. $n = 5$ trials for each network. (g) Raster plots of spontaneous activity and examples of inter-network SEs that spikes propagate from N_1 to N_2 (arrowhead) and from N_2 to N_1 (arrow). Red timestamps represent spikes participated in SEs. Scale bar: 10 sec and 500 ms. (h) Network-to-network influence estimated by thermoplasmonic neuromodulation. (i) Association degree in inter-network SEs. Red is the percentage of spikes participated in SEs. (j) Proportion of propagation direction within individual SEs. 106 detected SEs were analyzed.

Figure 6. Thermoplasmonic neuromodulation of a three-module network. (a) Phase-contrast image of tested networks. Agarose micropatterns were 500 μm by 500 μm with spacing of 100 μm . The asterisks indicate recording electrodes. Scale bar: 50 μm . (b) Raster plots of spontaneous activity from thirteen electrodes and examples of inter-network SEs that spikes propagate from $N_3 \rightarrow N_2$ (arrowhead) or from $N_3 \rightarrow N_2 \rightarrow N_1$ (arrow). Red timestamps represent spikes participated in SEs. Scale bar: 10 sec and 200 ms. (c) Correlation matrix at 38 DIV. (d) Association degree in inter-network SEs. Red is the percentage of spikes participated in SEs. (e) Proportion of propagation direction within individual SEs. 552 detected SEs were analyzed. (f) Spike rate change against power density for TP-NIR N_1 , TP-NIR N_2 , and TP-NIR N_3 . Black line (Ctrl) shows rate change in control groups (3 electrodes in unconnected networks) in the same MEA. $n = 5$ trials for each network. (g) Network-to-network influence estimated by thermoplasmonic neuromodulation. (h) Representative PSTHs for an electrical stimulation of a single electrode in each network. Bin size: 1 ms. (i) Mean PSTHs of all recording electrodes except stimulated ones. Stimulation electrodes: Ch3 and Ch5 in N_1 , Ch6 and Ch9 in N_2 , Ch12 and Ch13 in N_3 . 20 trials for each electrode. Bin size: 5 ms.

In summary, I find the novelty from the application perspective too limited in its current version. I encourage the authors, however, to address my concerns and provide a detailed roadmap & evidence how to build neural networks in situ using the approach provided i.e. in Figure 3.

>> Thank you for expressing your concern. We have added another data set (Figure 6) that used three neuronal networks and constructed a modular network using multiple connection lines, and made extensive revision in discussion section. We believe that this will be a sufficient demonstration of the application of our technology in building neural networks *in situ*.

We also added the following perspectives in discussion section,

“From the technological point of view, the technique of in situ manipulation using our platform was capable of implementing an in vitro experimental model that mimics the modular architecture often found in the brain. Although in this work we showed most of demonstrations using two networks, the proposed platform can construct various sizes and numbers of modules with strong intra-connections, and can generate network-to-network connections between network modules at a desired time window and locations. Moreover, it would be possible to directly investigate the causal relationship between the induced physical connections and the emergence of new global activity by selectively removing the connections between modules or by reversibly suppressing the module activity. In addition to the inhibitory effect through thermoplasmonic stimulation, it is expected that advanced analysis can be performed by incorporating diverse modulations through electrical or chemical stimulation. Furthermore, it will also be a powerful tool by combing our technique with the latest MEA technology that enable mapping synaptic connectivity using intracellular recording.³⁶”

< Minor corrections >

Some sentences were rephrased to clarify the meaning:

1. Result section. Effect of new connections on network synchronization

“ ... After creating new neurite connections, the activities of connected networks were investigated. Figure 3c shows **representative neuronal signals recorded before and after the ablation from four electrodes (Figure 3a, Ch1 and C2 in N₁, Ch3 and Ch4 in N₂) and Figure 3d presents raster plots of spike trains before the ablation and 3, 5, and 7 days after the ablation. After connecting two networks (N₁ and N₂), synchronized firing events between two networks emerged. ...**”

2. Result section. Thermoplasmonic neural ablation and network desynchronization

“Next, we attempted neurite ablation using thermoplasmonic stimulation and applied this technique to change the network structure by selectively removing of biological connections between networks. First, we examined the morphological effects of thermoplasmonic neurite ablation on neurites using optical micrographs. To perform the micro-ablation of neurites, **neurite outgrowth was induced in interconnection lines at 6 DIV using the same method from previous section. After 7 days of neurite outgrowth, NIR laser was focused on the center of neurites to induce thermoplasmonic neurite ablation** (Figure 4a, white dashed circles). The focused NIR laser (785 nm, beam diameter: 20.5 μm) with 4, 30, 68, and 101 W/mm^2 (1, 10, 22, and 33 mW) was illuminated for 5 seconds. This power range was determined based on a previous study on the effect of thermoplasmonic laser power on neural spiking activity, which showed irreversible loss of neural activity at 60.8 W/mm^2 (19 mW).²⁴ **At the power intensities of 4 and 30 W/mm^2 , there was little change of neurite morphology (Figure 4a). At power densities of 68 and 101 W/mm^2 , however, morphological neurite alterations occurred in and around the NIR-focused region. The observed morphological changes were neurite transection, beading, and fragmentation, which were used as indicators of neurite injury and degeneration in cultured nerve cells.²⁵⁻²⁷ At the highest power level of 101 W/mm^2 , it was found that the surrounding agarose hydrogel also melted due to the thermoplasmonic heat.** This strongly implied that the observed neurite damage and ablation were due to thermoplasmonic heating. The fluence of laser irradiation for neurite ablation (68 – 101 W/mm^2 for 5 seconds, 340 – 505 J/mm^2) was higher than the previous studies (24 – 48²⁸ or 60 – 720²⁹ or 18 – 54 J/mm^2 ³⁰), in which *in vitro* photothermal ablation of cancer cells **done using GNR membrane binding or uptake**. Another interesting observation was that there was extended outgrowth of neurites over the ablated areas after 4 days from the neurite ablation. **It can be inferred that even after removing the connections between the networks through the thermoplasmonic ablation, neurites can continue to grow out from the damaged or non-damaged neurons, thereby creating new connections again.** Thus, it was found that thermoplasmonic stimulation could temporarily induce micro-sized local damage of neurites for structural change, and neurites could regrow over the stimulated area. ...”

3. Rephrased wording are marked red in main text.

Reviewers' Comments:

Reviewer #2:

Remarks to the Author:

After reading the revised manuscript, I see that my issues have been addressed, in particular, in that more data have been added, which I consider very important.

In Figures 5 and 6, the recording channel/electrode numbers should be clearly indicated, and the electrodes should be labeled with the corresponding numbers in panels 5b and 6a.

Fig. 3f: Why are the fluctuations in the intra CC of the different cultures fairly correlated over the DIVs, if I look to the ups and downs in the fluctuation margins represented by the red area?

Fig. 3f: What happened to one of the cultures (blue line, trapezoid shape) in which the correlation went down to zero?

The manuscript still needs major language polishing, as it includes a multitude of grammatical and language errors.

Reviewer #3:

Remarks to the Author:

The authors carefully revised the manuscript and addressed all my concerns and comments in an appropriate way. There is only one minor rephrasing, I would suggest.

Instead of writing "thermoplasmonic neural stimulation can implement reversible neural suppression of the desired scale" I would specify that "thermoplasmonic neural stimulation can reversibly suppress neural activity of the desired scale".

Congratulations to the great work !

Reply to referees' comments

< Reviewer: 2 >

After reading the revised manuscript, I see that my issues have been addressed, in particular, in that more data have been added, which I consider very important.

In Figures 5 and 6, the recording channel/electrode numbers should be clearly indicated, and the electrodes should be labeled with the corresponding numbers in panels 5b and 6a.

>> Thank you for your comment. In Figure 5b, the electrode numbers (Ch1 and Ch2) were already indicated. As the reviewer pointed out, we have marked the electrode numbers on an MEA layout in Figure 6a.

Figure 5. Thermoplasmonic neuromodulation and estimation of network connectivity. ... (b) Phase-contrast image of tested networks. Agarose micropatterns were 500 μm by 500 μm with spacing of 100 μm . Scale bar: 50 μm .

Figure 6. Thermoplasmonic neuromodulation of a three-module network. (a) Phase-contrast image of tested networks. Agarose micropatterns were 500 μm by 500 μm with spacing of 100 μm . The asterisks indicate recording electrodes and the corresponding numbers are marked on an MEA layout. Scale bar: 50 μm .

Fig. 3f: Why are the fluctuations in the intra-CC of the different cultures fairly correlated over the DIVs, if I look to the ups and downs in the fluctuation margins represented by the red area?

>> As shown in a figure below, the fluctuation patterns in intra-CCs of four MEAs were different. What we wanted to show in Figure 3f was that the intra-CCs of four different cultures were distributed in the shaded region. One of the cultures, indicated with green line with square symbol, showed consistently high intra-CC values during the entire recording time, whereas the remaining cultures had fluctuations in their intra-CC values. This difference caused a fixed wide SD range, and it made all the intra-CCs of four cultures seem to fluctuate together. We will provide the mean values of intra-CCs for each MEA in a Source Data file.

Fig. 3f: What happened to one of the cultures (blue line, trapezoid shape) in which the correlation went down to zero?

>> In case of the culture, after 6 days, Ch1 and Ch2 in N1, and Ch3 in N2 showed a synchronized firing pattern (mean inter-CC: 0.482). After 7 days, there was a high tonic spiking activity in Ch3, which caused the a sharp decrease in the inter-correlation coefficient (mean inter-CC: 0.028).

<(Left) Phase-contrast micrographs of the culture (blue line, square symbol) six and seven days after agarose ablation. (Right) Raster plots of same days. N₁: top compartment, N₂: bottom compartment>

The manuscript still needs major language polishing, as it includes a multitude of grammatical and language errors.

>> In response to the reviewer's comment, we have done some polishing and the changes are marked as red in the new manuscript.

< Reviewer: 3 >

The authors carefully revised the manuscript and addressed all my concerns and comments in an appropriate way. There is only one minor rephrasing, I would suggest.

Instead of writing "thermoplasmonic neural stimulation can implement reversible neural suppression of the desired scale" I would specify that "thermoplasmonic neural stimulation can reversibly suppress neural activity of the desired scale".

Congratulations to the great work !

>> In response to the reviewer's comment, the corresponding sentence was revised.

In result section 'Network mapping by thermoplasmonic neuromodulation'

“Unlike electrical stimulation or optogenetics, thermoplasmonic neural stimulation can **reversibly suppress neural activity** at a desired scale by controlling the light power and does not require genetic modification, which would be beneficial for mapping network connectivity in situ.”